# Thermal tolerance of perovskite quantum dots dependent on A-site cation and surface ligand

Shuo Wang[1], Qian Zhao [1] ✉, Abhijit Hazarika [2,3], Simiao Li[1], Yue Wu[1], Yaxin Zhai [3,4], Xihan Chen [5], Joseph M. Luther [3] & Guoran Li [1] ✉

A detailed picture of temperature dependent behavior of $Cs_xFA_{1-x}PbI_3$ perovskite quantum dots across the composition range is constructed by performing in situ optical spectroscopic and structural measurements, supported by theoretical calculations that focus on the relation between A-site chemical composition and surface ligand binding. The thermal degradation mechanism depends not only on the exact chemical composition, but also on the ligand binding energy. The thermal degradation of Cs-rich perovskite quantum dots is induced by a phase transition from black γ-phase to yellow δ-phase, while FA-rich perovskite quantum dots with higher ligand binding energy directly decompose into $PbI_2$. Quantum dot growth to form large bulk size grain is observed for all $Cs_xFA_{1-x}PbI_3$ perovskite quantum dots at elevated temperatures. In addition, FA-rich quantum dots possess stronger electron−longitudinal optical phonon coupling, suggesting that photogenerated excitons in FA-rich quantum dots have higher probability to be dissociated by phonon scattering compared to Cs-rich quantum dots.

The unique and outstanding physicochemical properties of metal halide perovskites ($ABX_3$) make them promising materials for optoelectronic applications[1,2]. Quantum dots (QDs) of these materials offer band gap tunability and surface manipulation in addition to their exceptional bulk properties such as high defect tolerance and strong optical absorption and emission[3,4]. These features have enabled perovskite quantum dots (PQDs) to excel in various fields[5–7]. For instance, by passivating $CsPbX_3$ (X=Br, I, Cl) QD surfaces and replacing ligands, the photoluminescence quantum yield (PLQY) of PQDs has reached 90% in a wide range of 400–700 nm, and the external quantum efficiency (EQE) of red and green PQD light-emitting diode (LED) has both surpassed 20%[8–11]. PQD-based light-emitting memory and X-ray detector have also been realized with $CsPbBr_3$ PQDs[12,13]. Mixed A-site

$Cs_xFA_{1-x}$ PQDs particularly for photovoltaic (PV) applications have recently made a series of breakthroughs towards highly efficient low-cost solar cells and surpassed other QD systems in terms of efficiency, highlighting the potential in the development of commercial PQD devices for various applications[14–16].

At the present time, when talking about scale-up and commercialization of perovskite materials, one of the most critical obstacles is their poor stability under ambient environmental conditions[17]. The high surface area-to-volume ratio and complex surface chemistry of PQDs are very likely to bring more susceptibility to degradation induced by exposure to thermal stress, ambient air or chemicals. This also poses challenges to understand the distinctive degradation mechanisms of these nanoscale perovskites, especially for the PQDs

[1]Institute of New Energy Material Chemistry, School of Materials Science and Engineering, Nankai University, Tianjin 300350, China. [2]Polymers and Functional Materials Division, CSIR-Indian Institute of Chemical Technology, Uppal Road, Tarnaka, Hyderabad 500007, India. [3]National Renewable Energy Laboratory, Golden, CO 80401, USA. [4]Key Laboratory of Low-Dimensional Quantum Structures and Quantum Control of Ministry of Education, Department of Physics, Hunan Normal University, Changsha, Hunan 410081, China. [5]SUSTech Energy Institute for Carbon Neutrality, Department of Mechanical and Energy Engineering, Southern University of Science and Technology, Shenzhen, Guangdong 518055, China. ✉e-mail: qian.zhao@nankai.edu.cn; guoranli@nankai.edu.cn

such as FAPbI$_3$ and CsPbI$_3$ possessing suitable bandgap energies for solar cells. It has been found that the decomposition of CsPbI$_3$ PQDs does not occur in a pure oxygen atmosphere until light and water are involved[18]. Recently, Moot et al. further revealed that the compositional instability is attributed to reactive oxygen species initiated from oxygen, light and surface defect states, clearly demonstrating that the photo-oxidation degradation of CsPbI$_3$ PQDs is two orders of magnitude slower as compared to their thin-film or bulk counterparts, owing to the unique surface chemistries of PQDs[19]. Fortunately, the stability toward moisture, air and light can be dramatically improved by encapsulation and through the use of UV filters[20,21]. But the degradation caused by thermal stress seems inevitable since QD photovoltaic (PV) processing and operation require elevated temperatures[22–24], making it of fundamental importance to investigate detailed thermal behavior of PQDs[25].

Recent work by Boote et al. shows with the help of X-ray diffraction (XRD) and thermogravimetric analysis (TGA) that CsPbI$_3$ PQDs possess better photostability but worse thermal stability as compared to CsPbBr$_3$ PQDs, and also exhibit different degradation behavior from CsPbCl$_3$ PQDs[26]. They notice that the photostability of large CsPbI$_3$ PQDs is higher than that of small ones, proposing the stability difference may be related to surface chemistry[26]. As is well known, in addition to preventing moisture attack, organic ligands covering PQD surfaces induce the tensile lattice strain which not only stabilizes the black perovskite phase of CsPbI$_3$ at room temperature but also enables the full-range A-site tuning for Cs$_x$FA$_{1-x}$PbI$_3$[16]. Given the crucial role of surface ligands, more thermal studies are needed to understand the stability. Moreover, when we compare the thermal stability of PQDs with different compositions, the influence of ligand binding must be taken in account since the binding energy strongly depends on the surface chemical compositions. To the best of our knowledge, particularly for alloyed Cs$_x$FA$_{1-x}$PbI$_3$ PQDs of great potential, no such study has been conducted to date.

In this article, we establish the relation between the chemical composition and the surface ligand binding on thermal behaviors of Cs$_x$FA$_{1-x}$PbI$_3$ PQDs across the entire compositional range (x varied from 0 to 1) by systematically performing various measurements in-situ such as XRD, TGA and PL. We observe that FA-rich PQDs decompose into PbI$_2$ directly while Cs-rich PQDs first convert from black orthorhombic γ-phase to yellow orthorhombic δ-phase with increasing temperature. In contrast to their bulk counterparts, hybrid organic-inorganic FA-rich PQDs have slightly better thermal stability than all-inorganic CsPbI$_3$ PQDs. We have performed first principle density functional theory (DFT) calculations to study the strength of ligand binding (e.g., oleylamine and oleic acid) on the surface of Cs$_x$FA$_{1-x}$PbI$_3$ PQDs. The calculated bond strength of the ligand to FA-rich PQDs is larger than that for Cs-rich PQDs, illustrating the strong correlation between stability and ligand bond strength. The in situ PL studies show that longitudinal optical (LO) phonon-electron coupling is strengthened by increasing the FA/Cs ratio in PQDs, implying that the dissociation of photogenerated excitons by LO phonon scattering in FA-rich PQDs is easier when compared to Cs-rich PQDs.

## Results

To confirm the successful synthesis of Cs$_x$FA$_{1-x}$PbI$_3$ PQDs with entire compositional range, the corresponding optical and structural properties are demonstrated in Supplementary Fig. 1. The absorption onset and the PL emission peak position can be continuously tuned from ~650 nm for pure CsPbI$_3$ to ~800 nm for pure FAPbI$_3$ (Supplementary Fig. 1a and 1b). FA-rich PQDs (or pure FAPbI$_3$ PQDs) exhibit higher PLQY and longer time-resolved PL (TRPL) lifetime than Cs-rich PQDs (or pure CsPbI$_3$ PQDs), as demonstrated in Supplementary Fig. 1c and 1d. The XRD data show that all the prepared Cs$_x$FA$_{1-x}$PbI$_3$ PQDs exhibit black perovskite phase (cubic or near-cubic), and the detailed analysis on the tilting of the [PbI$_6$]$^{4-}$ octahedra in these PQDs can be found in

the previous work[27]. Here, in order to not increase the complexity of further analysis, we assume that Cs-rich PQDs are in γ-phase, and FA-rich as well as Cs$_{0.5}$FA$_{0.5}$PbI$_3$ PQDs are in α-phase. Since CsPbI$_3$ PQDs exhibit a smaller spacing of crystal layers than FAPbI$_3$ PQDs, the diffraction peak at ~27.7° shifts monotonically between the CsPbI$_3$ and FAPbI$_3$ PQD patterns (Supplementary Fig. 1e) following Bragg's and Vegard's laws. The $^1$H nuclear magnetic resonance (NMR) spectra demonstrate that the integral areas of both C-H (H1) and N-H (H2) in FA$^+$ cation linearly increase as the value of x in Cs$_x$FA$_{1-x}$PbI$_3$ decreases from 1 to 0 (Supplementary Fig. 2). The morphologies and size distributions of Cs$_x$FA$_{1-x}$PbI$_3$ PQDs are displayed in Supplementary Fig. 3, showing all PQDs retain the original cubic shape with a negligible mean size difference (<1 nm) after alloying. All above results are in good agreement with previous studies[14–16], which suggests the preparation method used here is reliable for achieving the full-range A-site tuning of Cs$_x$FA$_{1-x}$PbI$_3$ PQDs.

The thermal degradation mechanism for Cs$_x$FA$_{1-x}$PbI$_3$ PQDs with different compositions is first investigated by in situ XRD from 30 °C to 500 °C under argon flowing (Fig. 1). The detailed analyses of in situ XRD profiles are shown in Supplementary Fig. 4–8. For pure FAPbI$_3$ QDs, the major peaks of a cubic α-phase perovskite located at ~27.7° and ~31.0° (assigned to (002) and (012), respectively) can be seen clearly at initial temperature. When heating up to around 150 °C, the diffraction peaks at 25.2°, 29.0°, and 41.2°, attributed to (102), (103), and (006) planes of PbI$_2$, respectively, start appearing and the intensities increase with increasing temperature, which indicates that FAPbI$_3$ PQDs start to decompose into PbI$_2$. But, in the meantime, the peaks from the black phase perovskite surprisingly become stronger and sharper as the temperature increases from 150 °C to 300 °C. Thus, it allows us to state that part of FAPbI$_3$ PQDs which have not been decomposed into PbI$_2$ undergo grain growth and/or merging in this stage. The black-phase perovskite is completely decomposed into PbI$_2$ at 350 °C. Then the PbI$_2$ starts to melt and evaporate under the argon flow. Above 400 °C, only the peaks from the Pt substrate can be observed at 31.3°, 35.8°, and 39.5°. Based on the in situ XRD analysis, we conclude that FAPbI$_3$ PQDs directly degrade into PbI$_2$ as the heating temperature increases, and no observable perovskite phase transition occurs during thermal annealing. To exclude the possibility that the substrate type affects the results, we carried out ex situ XRD for FAPbI$_3$ PQDs deposited on various substrates (Supplementary Fig. 9). The obtained results are consistent with the in situ observations of thermal degradation of FAPbI$_3$ PQDs. It also should be noted that PbI$_2$ is the only condensed phase degradation product and the signals of FA-related decomposition products are not detected by XRD, which demonstrates FAI must leave the PQD sample under heating and is likely decomposed into gaseous products of HCN and NH$_3$[28].

Interestingly, as for the alloyed Cs$_x$FA$_{1-x}$PbI$_3$ PQDs, we notice that the peaks at 25.4°, 25.8°, 30.7° and 36.9° significantly increase when the degradation of the initial black-phase perovskite begins (Fig. 1a). These peaks can be assigned to the diffraction from the (105), (212), (015) and (020) planes of δ-phase perovskite, which indicates organic−inorganic hybrid Cs$_x$FA$_{1-x}$PbI$_3$ PQDs exhibit a phase transition with increasing temperature from black phase (corner-sharing octahedra) to yellow δ-phase (face-sharing octahedra), along with the decomposition into PbI$_2$. More interestingly, for FA-rich PQDs the decomposition reaction (i.e., appearance of PbI$_2$ phase) starts at ~150 °C as the first step, and the conversion to δ-phase concurrently occurs at ~180 °C as the subsequent step during thermal degradation process. However, for Cs-rich Cs$_x$FA$_{1-x}$PbI$_3$ PQDs, the black-to-yellow phase transition takes place before the decomposition into PbI$_2$. When we examine the in situ XRD data of pure CsPbI$_3$ PQDs, only a few characteristic diffraction peaks of PbI$_2$ can be observed during the whole period of the heating process. The degradation of CsPbI$_3$ PQDs under thermal stress could be mainly attributed to the phase transition from black γ-phase (~30–130 °C) to yellow δ-phase (~130–300 °C), and then finally to black

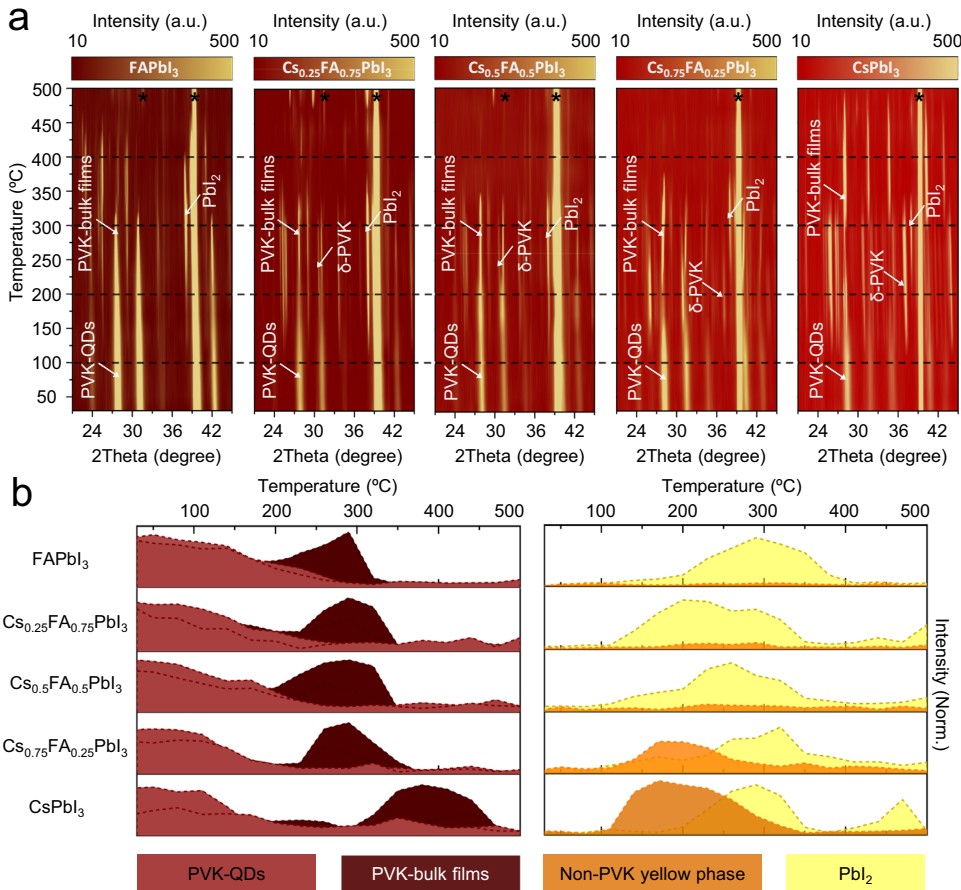

**Fig. 1 | In situ XRD analyses of $Cs_xFA_{1-x}PbI_3$ PQDs. a** In situ XRD patterns collected from 30 °C to 500 °C under argon flowing for $FAPbI_3$, $Cs_{0.25}FA_{0.75}PbI_3$, $Cs_{0.5}FA_{0.5}PbI_3$, $Cs_{0.75}FA_{0.25}PbI_3$, and $CsPbI_3$ PQDs. **b** Scheme for the dominant reflections in the thermal degradation of $Cs_xFA_{1-x}PbI_3$ PQDs that are drawn from the in situ XRD data taken at the $2\theta$ angles shown in Supplementary Table 1 for perovskite in QD form (PVK-QDs, red color shading), perovskite in bulk form (PVK-bulk films, solid dark-red color shading), orthorhombic non-perovskite (non-PVK yellow phase, orange color shading), and $PbI_2$ (yellow color shading). The peak of substrate is marked with asterisks. Source data are provided as a Source Data file.

α-phase (~300−450 °C) of perovskite. In addition, the sharpening and strengthening of the black α-phase perovskite peaks in the range of ~250−350 °C may be not only related to the grain growth, but also ascribed to the yellow-to-black phase transition. The XRD patterns and optical images of $CsPbI_3$ PQDs for each phase transition stage clearly shows that the initial reddish-brown, yellow and black $CsPbI_3$ films feature a γ-phase, a δ-phase and a α-phase, respectively (Supplementary Fig. 10). The difference in thermal degradation between pure $CsPbI_3$ and $FAPbI_3$ PQDs is also presented by UV−Vis spectra (Supplementary Fig. 11). It is to be noted that in the case of bulk $CsPbI_3$, different crystal phase transitions occur at different temperatures, but this is different from that of $CsPbI_3$ PQDs. For bulk $CsPbI_3$, yellow δ-phase at room temperature transforms into the black α-phase when heated to 360 °C. This black α-phase undergoes phase transformation into black β-phase at 260 °C and black γ-phase at 175 °C. These two kinetically stable black phases eventually turn back into the yellow δ-phase while cooled down[29]. For $CsPbI_3$ PQDs, the black phase can be stabilized at room temperature by bonding with surface ligands[30,31]. When heating up to ~130 °C, the surface ligands are evaporated, and the black γ-phase of $CsPbI_3$ PQDs is changed to the orthorhombic yellow phase which is more thermodynamically stable at this temperature, as described above.

In addition, gradual shifts of the perovskite peaks with increasing temperature are found in the XRD patterns, which can be ascribed to thermal expansion of the perovskite lattice[32]. The linear thermal expansion coefficients for different $Cs_xFA_{1-x}PbI_3$ PQDs are calculated following Eq. 1.

$$\alpha_L = \frac{1}{L_0} \times \frac{\partial L}{\partial T} \qquad (1)$$

where $\alpha_L$ is the linear thermal expansion coefficient and $L_0$ is the original length. The fitting details for the (002) peak (for cubic phase) are presented in Supplementary Fig. 12. The calculated values are $3.29 \times 10^{-5}\,K^{-1}$, $3.65 \times 10^{-5}\,K^{-1}$, $4.11 \times 10^{-5}\,K^{-1}$, $4.37 \times 10^{-5}\,K^{-1}$, and $4.53 \times 10^{-5}\,K^{-1}$ for $FAPbI_3$, $Cs_{0.25}FA_{0.75}PbI_3$, $Cs_{0.5}FA_{0.5}PbI_3$, $Cs_{0.75}FA_{0.25}PbI_3$, and $CsPbI_3$ PQDs, respectively. It can be seen that the linear thermal expansion coefficient increases with increasing the Cs content of PQDs. Although it is typically believed that the thermal expansion coefficient of hybrid inorganic-organic perovskites is larger than that of all-inorganic perovskites[33], this opposite trend for QD-form perovskite here is likely due to the significant negative surface energy and the lattice strain in Cs-based PQDs[31]. Moreover, to broaden the understanding of the degradation mechanism of these PQDs, we further perform in situ XRD of $Cs_xFA_{1-x}PbI_3$ PQDs, when the temperature is cooled down from ~200 °C (phase transition or decomposition temperature) to room temperature, as demonstrated in Supplementary Fig. 13. It can be seen that there is no phase transformation and degradation during the cooling down process.

To give clear indications and comparisons of thermal degradation processes of $Cs_xFA_{1-x}PbI_3$ PQDs with different stoichiometries, we

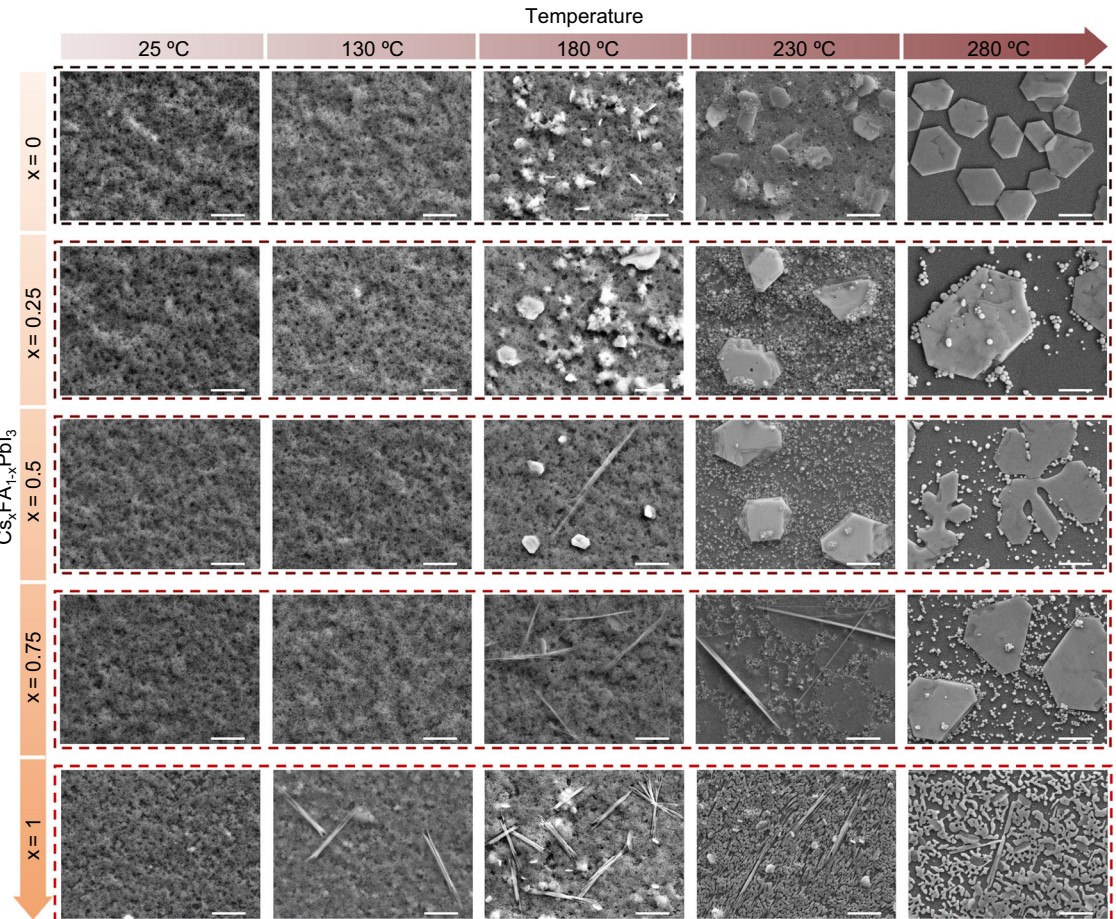

**Fig. 2 | Morphology evolutions of $Cs_xFA_{1-x}PbI_3$ PQDs during thermal degradation process.** SEM images of $FAPbI_3$, $Cs_{0.25}FA_{0.75}PbI_3$, $Cs_{0.5}FA_{0.5}PbI_3$, $Cs_{0.75}FA_{0.25}PbI_3$, and $CsPbI_3$ PQDs in thermal degradation process. Scale bar in each image is 4 μm.

present the dominant reflections of each component from the 2-dimensional XRD data in Fig. 1b and combine their corresponding morphology data shown in Fig. 2. The $2\theta$ angles used in Fig. 1b for $Cs_xFA_{1-x}PbI_3$ PQDs are presented in Supplementary Table 1. As illustrated in the left panel of Fig. 1b, the FA-rich PQDs are stable in the black perovskite phase in a wider temperature range without transforming into the yellow phase, decomposing into $PbI_2$ or transforming into bulk. However, all compositions of $Cs_xFA_{1-x}PbI_3$ PQDs eventually transform into the perovskite bulks above 200 °C. Pure $CsPbI_3$, once transformed into bulk, show better thermal stability in the black perovskite phase. In the case of FA-containing PQDs, the organic FA cation evaporates during the heating process, leaving behind only $PbI_2$. From the right panel of Fig. 1b, it can be seen that when $Cs_xFA_{1-x}PbI_3$ PQDs contain more Cs, the degradation process of PQDs is more dominated by phase transition from black perovskite to yellow non-perovskite phase. Pure $CsPbI_3$ PQDs exhibit a thermal degradation induced by the phase transition, while pure $FAPbI_3$ PQDs prefer to be directly decomposed to $PbI_2$. In other words, Cs-rich PQDs exhibit similar or even slightly poorer thermal stability of initial black perovskite phase than FA-rich PQDs, owing to the black-to-yellow phase transition at relatively low temperature. It also should be noted that no phase segregation is observed for all $Cs_xFA_{1-x}PbI_3$ PQDs from 25 °C to 150 °C. (Supplementary Fig. 14) Based on the analyses of morphological evolution of $Cs_xFA_{1-x}PbI_3$ PQDs during the thermal degradation process, the transformation to δ-phase nanorods is first noticeable in $CsPbI_3$ PQDs at 130 °C, and these ~8 μm nanorods gradually turn into ~2 μm bulk-like perovskite grains as the temperature is increased to 280 °C. With increasing FA, the dormant decomposition product becomes

hexagonal $PbI_2$, which is in good agreement with the XRD results. In the case of $Cs_{0.5}FA_{0.5}PbI_3$ PQDs, both the δ-phase nanorods and hexagonal $PbI_2$ are clearly noticeable in SEM images. Combined with the XRD result presented in Fig. 1, it allows us to assume that the degradation process of $Cs_{0.5}FA_{0.5}PbI_3$ PQDs seems to be governed by the decomposition into $PbI_2$, while the phase transition to yellow non-perovskite phase also plays a big role. These two trends are in competition to degrade $Cs_{0.5}FA_{0.5}PbI_3$ PQDs. In addition, we must note that here the amount of degradation products displayed in SEM images and the intensities of XRD peaks without any normalization are closely related to how many PQDs are in the scanning area during characterization (Supplementary Fig. 15), which therefore cannot represent the stability level of PQDs since the position of the scanning area is chosen randomly. Moreover, based on the SEM and XRD results, it is difficult to identify which type of PQDs have apparently higher thermal stability than the others.

It is well known that the thermal stability of all-inorganic perovskite bulk materials towards retaining desirable crystal phase (black phase) for device applications as light harvesting materials, is much higher than that of organic-inorganic hybrid halide perovskites[34,35], but which stands in contrast to the case in their nanocrystal counterparts as discussed above. Furthermore, pure $CsPbI_3$ and $FAPbI_3$ PQDs both display poorer thermal stability when compared to their bulk counterparts. Likewise, Boote and Liao et al. show that pure $CsPbI_3$ PQDs undergo a thermal degradation from the luminescent γ-phase to a combination of $PbI_2$ and δ-phase $CsPbI_3$ beginning at around 100 °C, which is much lower than that for $CsPbI_3$ thin films/bulks whose black-to-yellow phase transition occurs above 200 °C[26,36,37]. These unique

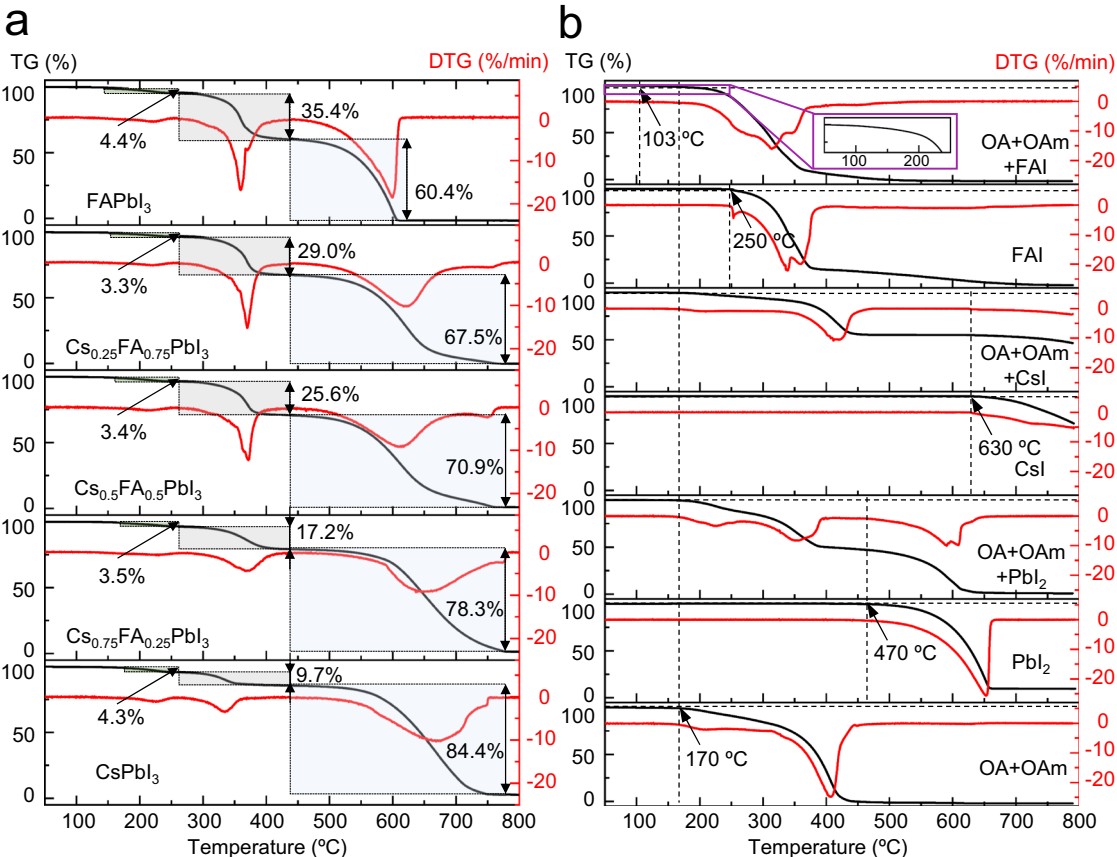

**Fig. 3 | Mass loss for thermal degradation analyses of $Cs_xFA_{1-x}PbI_3$ PQDs. a** TG and DTG curves for $Cs_xFA_{1-x}PbI_3$ PQDs with different FA/Cs ratios. The shading represents the percentage of mass loss in the relevant temperature range. **b** TG and DTG curves for the pure FAI, CsI, $PbI_2$ and the mixtures of OA+OAm, CsI+OA+OAm, FAI+OA+OAm, and $PbI_2$+OA+OAm. The temperature is increased from room temperature to 800 °C in argon environment. The inset is the enlarged part of the data obtained from 50 °C to 250 °C for FAI+OA+OAm mixture. Source data are provided as a Source Data file.

and opposite trends in PQDs must be attributed to a synergistic effect of chemical composition and surface energy induced by ligand binding, which is the key to stabilize black perovskite phase at room temperature especially for Cs-alloyed perovskite nanoparticles[31].

Simultaneous thermogravimetry analysis (TGA) with derivative thermogravimetry (DTG) is performed to determine the mass loss of $Cs_xFA_{1-x}PbI_3$ PQDs with different FA/Cs ratios during the heating process, aiming at understanding the thermal degradation mechanism in the presence of organic ligands. As demonstrated in Fig. 3a, the mass loss of all the PQD samples proceeds as a three-step process under flowing argon. The first mass loss reflects a combination of the degradation of perovskite black phase and the removal of organics such as OA, OAm and possibly residual octadecene that is in excess during the PQD synthesis. A more significant secondary mass loss occurs above 260 °C, arising from the remaining surface ligands (OA/OAm) and the solely inside organic component (FAI) in $Cs_xFA_{1-x}PbI_3$ PQDs[27]. The major third process above 430 °C could be assigned to the loss of $PbI_2$, probably coinciding with the formation of the other decomposition products like CsI that is similar to recent reports showing TGA curves of $CsPbI_3$[26,37].

It can be found that no significant differences are observed in the onset temperature (~150 °C) for all $Cs_xFA_{1-x}PbI_3$ PQDs, which may be due to the lack of TGA measurement accuracy and precision. But it is obvious that the initial mass loss temperatures of PQDs are much lower than that of the bulk $CsPbI_3$ and $FAPbI_3$ which begin to degrade above 450 °C and 300 °C, respectively[28,37]. To further probe the effect of organic ligands on mass changes of PQDs during TGA measurements, we plot the relative mass loss versus temperature for the pure FAI, CsI,

$PbI_2$ and their respective OA/OAm mixture such as CsI/OA/OAm, FAI/OA/OAm and $PbI_2$/OA/OAm in Fig. 3b. In comparison to the mass loss events for pure FAI (from ~250 °C to ~360 °C) and pure OA/OAm ligands (from ~170 °C to ~440 °C), an apparent temperature delay (around 67 °C) of the mass loss onset for OA/OAm/FAI mixture is observed at ~103 °C and indicative of the interaction between the FA cation and the OA/OAm ligand[28]. This interaction may lead to the extraction of FA cation from the perovskite structure as the organic capping ligands on the PQD surfaces are evaporated away, which can partially explain why removing the surface ligand is always accompanied by the decomposition in FA-based PQDs. In the case of pure $CsPbI_3$ PQDs, such temperature delay is negligible in TGA curves for CsI/OA/OAm and $PbI_2$/OA/OAm mixtures, suggesting that the CsI-OA-OAm and $PbI_2$-OA-OAm interactions are much weaker than the interaction between FAI and the capping ligands. However, the removal of organic ligands at the elevated temperature likely causes the instability of $CsPbI_3$ PQDs. The reason for this instability is that the surface strain imparted by the ligands to stabilize the black perovskite phase against phase degradation to thermodynamically favored δ-phase is unbalanced while leaving the $CsPbI_3$ PQD surface bare. Hence, regardless of FA-rich and Cs-rich PQDs, the thermal stability of these PQDs is firmly associated with their ligand bindings. Many recent studies demonstrate that the strong bonding between PQDs and the functional surface ligands such as 2,2-iminodibenzoic acid[38], hexamethyldisilathiane[39], trioctylphosphine[40], 1-octadecanethiol[41], 1-propyl-3-methylimidazolium iodide[42], 2-naphthalenesulfonic acid[43], sulfur−oleylamine[44], etc., is beneficial in protecting the phase stability of the PQDs. Particularly, Dutta et al. show that the introduction of

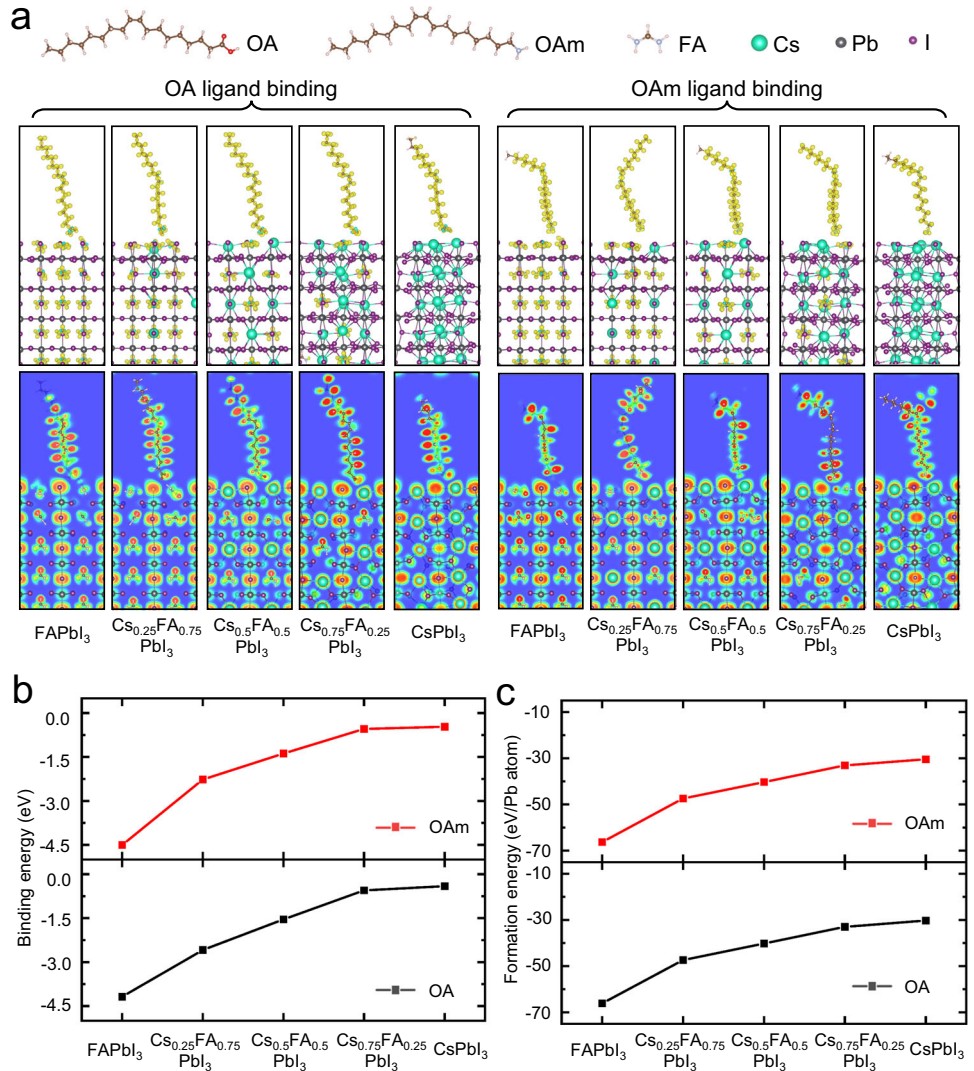

**Fig. 4 | Ligand binding properties for $Cs_xFA_{1-x}PbI_3$ PQDs of various compositions. a** Charge density difference plots (upper) and electron localization function analyses (below) for one OA/OAm ligand adsorbed on the surface of $Cs_xFA_{1-x}PbI_3$ PQDs. Red and green denotes areas of electron accumulation and depletion in electron localization function analysis, respectively. **b** DFT calculated binding energy of OA and OAm ligands on different $Cs_xFA_{1-x}PbI_3$ PQD surfaces. **c** The corresponding calculated formation energy of $Cs_xFA_{1-x}PbI_3$ PQDs after binding OA/OAm ligands. Source data are provided as a Source Data file.

separately mixed oleylamine and hydroiodic acid (OLA-HI) can lead to a substantial stabilization of $CsPbI_3$ PQDs at 260 °C (100 °C higher than the usual synthesis temperature) via enhancing the binding of oleylammonium ions on QD surface[45]. It should be noted that the discrepancy in the temperature points for the changes of PQD samples (such as the phase transitions, the emerging, melting and evaporation of $PbI_2$) between different measurement techniques may be ascribed to the differences in the heating process, the instrument parameters (accuracy rate, sensitivity and program system, etc.), as well as the amount and the exposed surface area of the PQD samples used in characterizations (Supplementary Fig. 15).

Thus, to better understand the binding nature of OA/OAm ligands on the PQD surface and its effect on thermal stability, we then perform self-consistent density functional theory (DFT) calculations for $Cs_xFA_{1-x}PbI_3$ PQDs of various compositions with capping ligands. According to previous research and our XRD results above, the FA-rich (in α-phase) and the Cs-rich (in γ-phase) PQD surfaces with similar structural features are modelled using the (010) and the (101) facet, respectively[31,46]. The charge density difference plots and electron localization function analyses for one OA/OAm ligand adsorbed on the surface of different $Cs_xFA_{1-x}PbI_3$ PQDs are shown in Fig. 4a. The DFT calculated binding energy of ligands and formation energy of $Cs_xFA_{1-x}PbI_3$ PQDs after ligand binding are presented in Fig. 4b, c, respectively. Tabulated values can be found in Supplementary Table 2. Specifically, the binding energies of ligands on a $(hkl)$ plane $(E_{b,(hkl):ligand})$ of PQDs are computed using slab geometry from the following equation as Eq. 2:

$$E_{b,(hkl):ligand} = \left( E_{(hkl)} + E_{ligand} \right) - E_{(hkl):ligand} \qquad (2)$$

where $(E_{(hkl)} + E_{ligand})$ and $E_{(hkl):ligand}$ represents the computed energies of the separated and adsorbed molecule on $(hkl)$ slab, respectively[47]. The formation energies $(E_{form.})$ of the ligand-capped $Cs_xFA_{1-x}PbI_3$ PQDs are calculated by Eq. 3:[48]

$$E_{form.} = E_{PQD,ligand} - n_{Cs} \times \mu_{Cs} - n_{Pb} \times \mu_{Pb} - n_I \times \mu_I - n_C \times \mu_C$$
$$- n_N \times \mu_N - n_O \times \mu_O - n_H \times \mu_H \qquad (3)$$

where $E_{PQD,ligand}$ is the total energy of PQDs after ligand binding, and $\mu_{Cs}$ $(n_{Cs})$, $\mu_{Pb}$ $(n_{Pb})$, $\mu_I$ $(n_I)$, $\mu_C$ $(n_C)$, $\mu_N$ $(n_N)$, $\mu_O$ $(n_O)$ and $\mu_H$ $(n_H)$ are the chemical potentials (numbers) of cesium, lead, iodine, carbon, nitrogen, oxygen and hydrogen, respectively.

As seen in the analysis of ligand binding properties for $Cs_xFA_{1-x}PbI_3$ PQDs of Fig. 4, the charge depletion and accumulation regions prominently involve the lone-pair interactions and polarizations, as well as the hydrogen bonds between (Cs−I terminated or FA−I terminated) PQD surfaces and the carboxylic group in OA or the amino group in OAm[49,50]. The surface-exposed FA cations in PQDs can bond to the carboxylic or amino groups of ligands via its extra hydrogen bonds, resulting in an increased ligand binding energy of both OA and OAm on FA-rich PQD surface as demonstrated in Fig. 4b, which may further lead to higher surface stability of FA-rich PQDs when compared to that for Cs-rich PQDs. As illustrated in Fig. 4c, FA-rich PQDs exhibit more negative values of formation energy than Cs-rich PQDs, suggesting that the FA-rich PQDs having the perovskite phase with corner shared octahedra is more thermodynamically favorable at room temperature.

To further clarify the relation between the chemical composition and surface ligand binding, and how it impacts the thermal stability, we prepared $Cs_{0.75}FA_{0.25}PbI_3$ PQDs with trioctylphosphine oxide (TOPO) as surface binding ligands, and performed XRD experiments as well as DFT theoretical calculations on them. In Supplementary Figs. 16–18, the $Cs_{0.75}FA_{0.25}PbI_3$ PQDs with TOPO ligands exhibit better thermal stability than the ones with OA and OAm ligands. The binding energy of TOPO ligands on $Cs_{0.75}FA_{0.25}PbI_3$ PQDs (−1.15 eV) is higher than that of OA (−0.55 eV) and OAm (−0.54 eV) ligands. These results suggest that the different ligands capping on the same kind of PQDs provide different thermal stabilities, through changing the binding between the ligands and the PQD surface. Combined with the results from the above XRD and DFT analyses on PQDs with same surface ligands but different chemical compositions, it implies that the phase stability is synergistically influenced by the chemical composition and the surface ligands.

Moreover, the temperature dependence of the optical characteristics in different $Cs_xFA_{1-x}PbI_3$ PQDs is probed by in situ PL spectroscopy shown in Fig. 5. The PL-temperature pseudo color maps demonstrate the position shift and broadening in PL peaks over a temperature range of 80–390 K, which tell us about not just stability but phonon scattering that limits charge-carrier mobilities and governs emission line broadening in PQDs[51]. (Fig. 5a) The corresponding 1D PL plots as function of temperature are presented in Supplementary Fig. 19. Specifically, the exciton-phonon interaction in each $Cs_xFA_{1-x}PbI_3$ PQDs is examined by plotting the full width at half-maximum (FWHM) of the PL spectra, along with best fits obtained via least square fitting using Segall's expression as Eq. 4:[52,53]

$$\Gamma(T) = \Gamma_0 + \Gamma_{AC} + \Gamma_{LO} + \Gamma_{imp} = \Gamma_0 + \gamma_{AC}T + \frac{\gamma_{LO}}{e^{(E_{LO}/k_BT)}-1} + \frac{\gamma_{imp}}{e^{E_{b,imp}/k_BT}} \tag{4}$$

where $\Gamma(T)$ is the overall PL emission broadening of the lowest-lying 1S exciton, $\Gamma_0$ is a temperature-independent inhomogeneous broadening term associated with exciton–exciton interactions, crystal disorder and imperfections. The second term ($\Gamma_{AC}$) represents a homogeneous broadening term mainly relating to acoustic phonon scattering, which is linearly dependent on temperature with an exciton-acoustic phonon coupling coefficient ($\gamma_{AC}$). The third term ($\Gamma_{LO}$) is a homogeneous broadening term that arises from LO phonon (Fröhlich) scattering, and follows the Bose−Einstein distribution function of the LO phonons given as $\Gamma_{LO} = \frac{\gamma_{LO}}{e^{(E_{LO}/k_BT)}-1}$, where $\gamma_{LO}$ is exciton-longitudinal optical phonon (Fröhlich) coupling coefficient, $E_{LO}$ is an energy representative of the frequency for the weakly dispersive LO phonon branch, $k_B$ is Boltzmann's constant and $T$ is the temperature. In the case of the presence of fully ionized impurities, an additional inhomogeneous broadening term ($\Gamma_{imp}$) can be added in Eq. 4, due to the phonon scattering from ionized the impurities as given by $\Gamma_{imp} = \frac{\gamma_{imp}}{e^{E_{b,imp}/k_BT}}$, where $E_{b,imp}$ is the average binding energy of impurities. We here

assume $\Gamma_{imp} \approx 0$ since impurity term could not produce the linear linewidth variation above the temperature of ~100 K as observed in Fig. 5b[54,55].

As the contribution of acoustic phonon for polar inorganic semiconductors is usually ignored in many studies[51,56], we first proceed to fit all data of $Cs_xFA_{1-x}PbI_3$ PQDs using the equation only including $\Gamma_0$ and $\Gamma_{LO}$ terms as shown in Supplementary Fig. 20. However, the $\Gamma_0$ and $\Gamma_{LO}$ term could not produce excellent fits in the temperature range of ~80−200 K, suggesting the acoustic phonon scattering may have some influence on PL broadening in the low-temperature range. Then, we incorporate $\Gamma_{AC}$ term in the phonon modes, yielding a much better fit than the previous method to the measured data as demonstrated in Fig. 5b. The parameters derived via both $\Gamma_{AC}$-free and $\Gamma_{AC}$-included fitting methods for each PQD type are presented in Supplementary Tables 3, 4. These resulting parameters from both fittings reveal that the majority of PL broadening in the high temperature regime is contributed from electron−LO phonons interactions, while acoustic phonon scattering governs the broadening for the low temperature ($T \leq 100$ K) due to the large reduction of LO phonon population. The tendency in $\Gamma_0$ is mainly related to the difference in chemical composition for $Cs_xFA_{1-x}PbI_3$ PQDs having similar size and shape (Supplementary Fig. 3). Another significant tendency is the increase in Fröhlich coupling branch with the increase of the FA amount in PQDs, demonstrating the higher LO phonon energy for FA-rich PQDs correlating with the smaller high-frequency value of the dielectric function and higher polarity than that for Cs-rich PQDs[51,57,58]. The exciton binding energies for $Cs_xFA_{1-x}PbI_3$ PQDs are estimated from Fig. 5c and Supplementary Fig. 21 following the equation given by Eq. 5:

$$I(T) = \frac{I_0}{1 + Ae^{-E_{b,ex}/k_BT}} \tag{5}$$

where $E_{b,ex}$ is the exciton binding energy, A is a constant, $I(T)$ and $I_0$ are the integrated PL intensities at temperature T and 0 K, respectively. Combined with the calculated exciton binding energies shown in Supplementary Table 5, it allows us to conclude that the high $E_{LO}$ and $\gamma_{LO}$ relative to $E_{b,ex}$ implies that the photogenerated excitons in alloyed FA-rich PQDs have higher probability to be disassociated by LO phonon scattering and thus are well suited for photovoltaics, photocatalysis and photodetector[59]. Although the exact values for the $E_{b,ex}$ in these PQD systems are still a matter of debate as illustrated in Supplementary Table 5 (obtained from different reported methods with a few to a few tens of milli-electron volts), the tendency of dissociation through one LO phonon in FA-rich PQDs seems to be sufficiently strong. Regarding to $\Gamma_{AC}$-included fitting results, the acoustic phonon coupling strength is increased by increasing the Cs/FA ratio in $Cs_xFA_{1-x}PbI_3$ PQDs, while pure $CsPbI_3$ PQDs show similar value of $\gamma_{AC}$ with pure $FAPbI_3$ PQDs. This is maybe owing to the Cs-alloying in $Cs_xFA_{1-x}PbI_3$ PQDs that brings additional lattice strain and defects via A-site cation migration accompanied with surface ligand dissociation and binding. In addition, the width narrowing in red colour at the temperature range of ~250−330 K (Fig. 5a) is a visual illusion of the decrease in PL FWHM which is actually caused by the decrease of PL intensity. It can be seen in Fig. 5c that an abrupt change in the PL intensity occurs at ~250 K and ~330 K. The PL emission intensity for most of semiconductors is usually observed to decrease monotonically with increasing temperature (termed as thermal quenching), basically due to the increase of the nonradiative recombination probability of electrons and holes. However, for these $Cs_xFA_{1-x}PbI_3$ PQDs, the increase of PL intensity in the range of ~250−330 K is attributed to that the electrons captured by defects gain energy from collisions or surroundings to escape from the defect cavity and participate in radiative recombination, overcoming the influence of the thermal quenching and giving

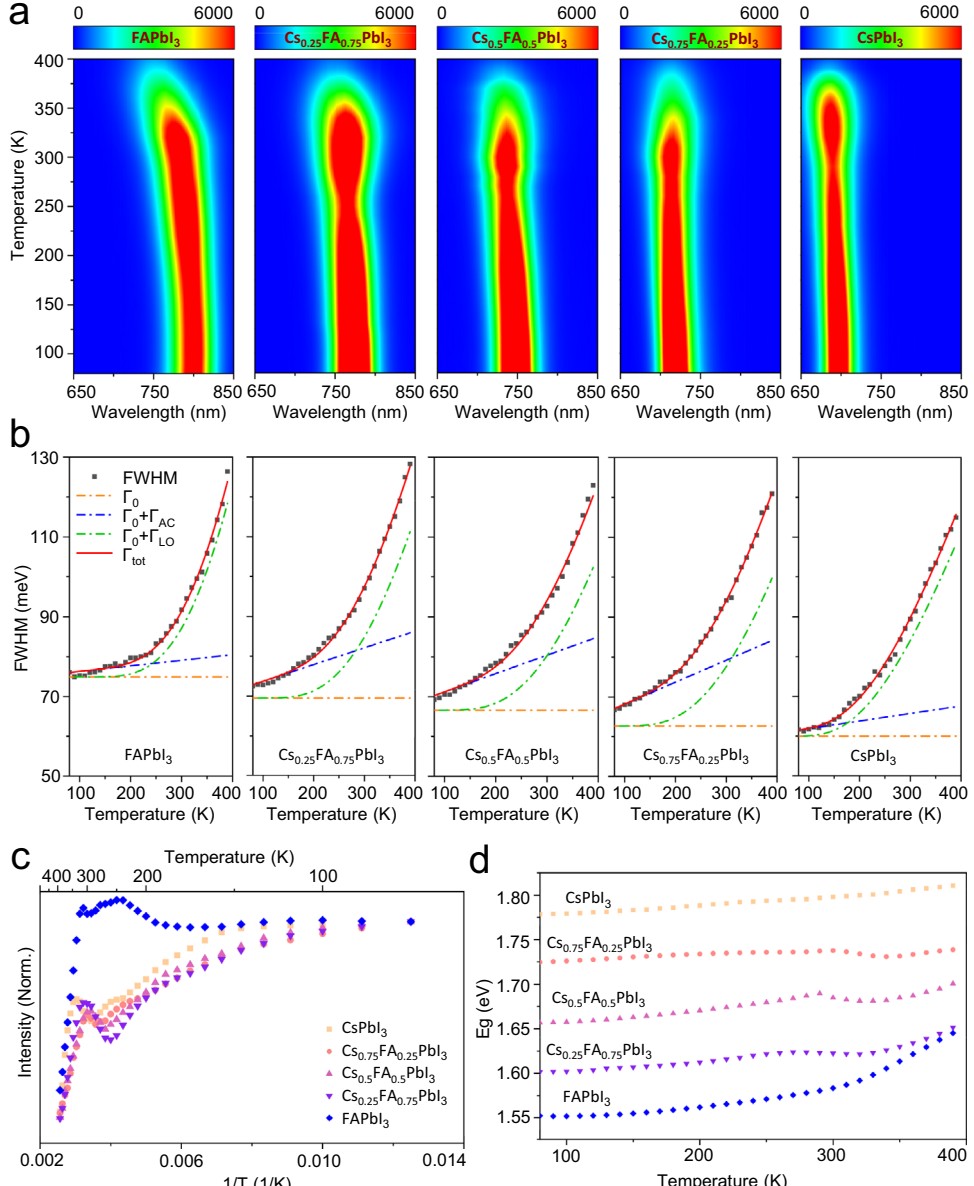

**Fig. 5 | Temperature dependent optical characteristics of $Cs_xFA_{1-x}PbI_3$ PQDs.**
**a** In situ PL spectra for $FAPbI_3$, $Cs_{0.25}FA_{0.75}PbI_3$, $Cs_{0.5}FA_{0.5}PbI_3$, $Cs_{0.75}FA_{0.25}PbI_3$, and $CsPbI_3$ PQDs. **b** FWHM extracted from in situ PL spectra. The data are normalized to the maximum intensity. The black squares are the experimental data fitted using Segall's expression (red solid line). The PL exciton linewidth broadening arises from inhomogeneous broadening term $\Gamma(T) = \Gamma_0$ (orange dash-dot line), acoustic phonon interaction $\Gamma(T) = \Gamma_0 + \Gamma_{AC}$ (blue dash-dot line), and longitudinal optical phonon interaction $\Gamma(T) = \Gamma_0 + \Gamma_{LO}$ (green dash-dot line). **c, d** The temperature dependence of normalized PL intensities and bandgaps for different $Cs_xFA_{1-x}PbI_3$ PQDs. The PL intensity is normalized by its value at 80K. The bandgap values are extracted using the Gauss fit. Source data are provided as a Source Data file.

rise to the PL intensity. It has been reported that this variation process in PL intensity is also dependent on the excitation power of PL measurements[56].

The perovskite bandgap energies as a function of temperature are heavily influenced by the crystal structure it adopts[60]. At the elevated temperature, the expanded perovskite lattice leads to a decreased overlap between the $I^-$ and $Pb^{2+}$ antibonding orbital, decreasing the valence band energy and consequently increasing the bandgap[61]. In the case of $Cs_xFA_{1-x}PbI_3$ PQDs, various degrees of lattice expansion can be caused by different A-site compositions as well as the ligand-induced strain. In Fig. 5d, all PQD types show an initial linear blue-shift below ~280 K and a more complex behavior above ~300 K. It is well known that the derivative of the bandgap ($E_g$) over temperature ($T$) under constant pressure ($P$) can be generally expressed using a quasi-

harmonic approximation as Eq. 6:[62]

$$
\begin{aligned}
\frac{dE_g}{dT} &= \left(\frac{\partial \ln V}{\partial T}\right)_P \left(\frac{\partial P}{\partial \ln V}\right)_T \left(\frac{\partial E_g}{\partial P}\right)_T + \sum_{j,\vec{q}} \left(\frac{\partial E_g}{\partial n_{j,\vec{q}}}\right)\left(n_{j,\vec{q}} + \frac{1}{2}\right) \\
&= -3\alpha_V B_0 \left(\frac{\partial E_g}{\partial P}\right)_T + \sum_{j,\vec{q}} \left(\frac{\partial E_g}{\partial n_{j,\vec{q}}}\right)\left(n_{j,\vec{q}} + \frac{1}{2}\right)
\end{aligned}
\tag{6}
$$

where $\alpha_V$ is the volumetric thermal expansion coefficient, $B_0 = -V\left(\frac{\partial P}{\partial V}\right)_T$ is the bulk modulus, and $n_{j,\vec{q}}$ represents the number of phonons at $j$ branch with wavevector of $\vec{q}$ following Bose–Einstein distribution for bosons given as $n_{j,\vec{q}} = \frac{1}{e^{\hbar\omega_{j,\vec{q}}/k_BT}-1}$, where $\omega_{j,\vec{q}}$ is the angular frequency of the phonon mode[54,63]. In Eq. 6, the first term

represents temperature-induced contraction/expansion of the PQD lattice (thermal expansion), and the second term is associated with the change of the electronic band structure by lattice vibrations (electron−phonon interaction). For typical halide perovskite materials, the $\left(\frac{\partial E_g}{\partial P}\right)_T$ in the first term which depends on the detailed electronic band structure and the bonding parameters is negative thus resulting in a positive relationship between temperature and bandgap[61,62,64]. The second part in Eq.6 generally consists of two terms: Debye-Waller term which describes the interactions between an electron and two phonons in the branch $j$ with the opposite wave vectors, and Fan term which describes the virtual phonon emission and absorption processes[65]. Although it is challenging to calculate the direct contribution of electron−phonon interaction with considering all possible phonon modes in entire Brillouin zone, the electron−phonon term, $\sum_{j,\vec{q}}\left(\frac{\partial E_g}{\partial n_{j,\vec{q}}}\right)\left(n_{j,\vec{q}}+\frac{1}{2}\right)$, can be expressed by using a two-oscillator model only involving two dominant phonon modes (acoustic phonon and optical phonon) as Eq. 7:[66]

$$\triangle E_g = -F_1\sum_{\vec{q}}\frac{4m^*|q|}{q^2-Q^2}\cdot\frac{1}{e^{hv_q/k_BT}-1} - F_2\sum_{\vec{q}}\frac{m^*q^2}{q^4-P^4}\cdot\frac{1}{e^{E_{LO}/k_BT}-1}$$

(7)

Where $q$ is the phonon wave vector, $v_q$ is the velocity of acoustic wave near the zone edge, $m^*$ is the effective mass of charge carriers, $Q = |\frac{2m^*v_q}{h}|$ for acoustic phonon modes, $P = \frac{\sqrt{2m^*E_{LO}}}{h}$ for LO phonon modes, $F_1$ and $F_2$ are positive constants. When we consider a condition so that $P < |q| < Q$ as generally reported in perovskite materials[54,55], then the acoustic phonon interaction tends to increase the bandgap, while optical phonon interaction results in decreasing the bandgap with increasing temperature[66].

Overall, it is clear that the increase of bandgap for $Cs_xFA_{1-x}PbI_3$ QDs is dominated by the combination of thermal expansion and electron−acoustic phonon interaction. The S-shape behavior shown in Fig. 5d for alloyed PQDs (250–350 K) may be ascribed to the significantly increased electron−LO phonon interaction in this temperature range. However, this S-shape behavior in bandgap variation can also be resulted from phase transition that is previously reported in perovskite bulks, demonstrating that FA-based halide perovskite exhibits a continuous phase transition from the cubic α-phase to the tetragonal β-phase at ~285 K[67]. It is difficult to accurately and precisely examine how each fraction affects PL emission in these $Cs_xFA_{1-x}PbI_3$ QD systems based on the current experimental data. Many studies have also shown the high complexity in the temperature dependence of the band-gap energies for these PQDs is owing to the unique ligand binding on the soft perovskite lattice accompanied with quantum confinement, which is distinguished from their bulk counterparts[54,68,69]. We may further inspect in more detail the temperature-dependent PL line shapes and carry out time-resolved measurements to find out the specific charge carrier cooling mechanisms with quantified analyses. We also believe that it will be an interesting research topic to examine the quantitative relationship between the surface ligand left in QD film and the start of grain growth during the heating process. Although such additional analyses are beyond the scope of the present investigation, we would like to stress here that fundamental studies focusing on $Cs_xFA_{1-x}PbI_3$ PQDs are urgently required and precise knowledge about the dependence of their properties on temperature is mandatory for numerous optoelectronic applications such as LEDs which when operated could create local heat, changing the emission color.

## Discussion

Together with theoretical calculations, the in situ probing of crystal structure, morphology, and optical properties provide a detailed picture of thermal behavior for $Cs_xFA_{1-x}PbI_3$ PQDs with a more complete understanding of the concurrent effects of QD chemical composition and surface ligand binding. The thermal degradation process of Cs-rich PQDs is dominated by a phase transition from black γ-phase (~30–130 °C) to yellow δ-phase (~130–300 °C) and then finally to black α-phase (~300–450 °C), while FA-rich PQDs are majorly decomposed into $PbI_2$, showing similar or even slightly higher thermal stability than Cs-rich PQDs. This difference of degradation behavior can be attributed to not only the variation of chemical composition, but also the extra hydrogen bonds formed between the surface-exposed FA cations of PQDs and the carboxylic or amino groups of ligands, which also leads to an increasing binding energy of OA/OAm ligands with the increase in the FA/Cs ratio. Moreover, all compositions of $Cs_xFA_{1-x}PbI_3$ PQDs exhibit grain growth to produce perovskite bulks at elevated temperature, as previously reported in other systems[70–72]. More interestingly, the photogenerated excitons in FA-rich PQDs have higher probability to be dissociated by LO phonon scattering than that in Cs-rich PQDs, which is favorable for photovoltaic applications. This work represents an initial but deep thermodynamic understanding of a complex new material, and is meant to serve as a guide for those seeking novel surface modifications for PQDs.

## Methods

### Materials

Oleylamine (OAm; technical grade, 80–90%) and 1-octadecene (1-ODE; technical grade, 90%) were purchased from Acros Organics. Cesium carbonate ($Cs_2CO_3$; 99.99%) and oleic acid (OA; technical grade, 90%) were purchased from Alfa Aesar and Sigma-Aldrich, respectively. $PbI_2$ (99.9%) and methyl acetate (MeOAc, anhydrous, 99.5%) were purchased from Advanced Election Technology CO. Ltd and Aladdin, respectively. Hexane (GC, ≥ 96%) and formamidinium acetate (FA-acetate, 99%) were purchased from TCI. All chemicals are used directly without further purification.

### CsPbI₃ PQD synthesis

$CsPbI_3$ QDs were synthesized via a hot-injection method. A Cs-oleate precursor was prepared by adding 0.407 g $Cs_2CO_3$, 20 mL 1-ODE and 1.25 mL OA into a three-necked round bottom flask and degassing it under vacuum at 80 °C for 30 min. The mixture was then heated to 130 °C under $N_2$ to obtain a clear Cs-oleate precursor for use in the next step of hot-injection. In another flask, 0.5 g $PbI_2$ and 25 mL 1-ODE were mixed and degassed under vacuum at 120 °C for 30 min. Then, a mixture of 2.5 mL OA and 2.5 mL OAm, preheated to 120 °C, was injected into the flask. After $PbI_2$ was fully dissolved, the reaction mixture was heated to 180 °C under $N_2$ flow. 2 mL of the Cs-oleate precursor solution was immediately injected into the $PbI_2$ solution, and then the mixtures was quenched by submersing the flask into an ice-water bath at 5 s after injection. After cooling to room temperature, the resultant solution was combined with 70 mL MeOAc and then centrifuged at 5970 × g for 5 min. The precipitate was dispersed in 5 mL hexane, and after adding 5 mL MeOAc it was centrifuged again at 5970 × g for 5 min. The final precipitate was dispersed in 5 mL hexane and stored in a refrigerator at 5 °C before use.

### FAPbI₃ PQD synthesis

$FAPbI_3$ QDs were synthesized via a hot-injection method. FA-oleate precursor was prepared by mixing 0.521 g FA-acetate with 10 mL OA and then degassing the mixture under vacuum at 60 °C for 30 min. The temperature was subsequently increased to 120 °C and a clear solution was obtained. After that, the mixture was cooled to 90 °C and maintained in $N_2$ for injection. For $PbI_2$ solution, a mixture of 0.344 g $PbI_2$ and 20 mL 1-ODE was degassed under vacuum at 120 °C for 30 min in a

three-necked round-bottom flask. Then, 5 mL OA and 3 mL OAm were heated to 120 °C and injected into the flask. After $PbI_2$ was fully dissolved, the temperature of the mixture was reduced to 80 °C under $N_2$. 5 mL FA-oleate precursor solution was swiftly injected into the $PbI_2$ mixture at 80 °C. After around 15 s, the reaction was quenched using an ice-water bath. When the mixture was cooled to room temperature, 9 mL MeOAc was added to the mixtures and centrifuged at $6790 \times g$ for 30 min. The residual precipitate was dispersed in 7 mL hexane, re-precipitated with 5 mL MeOAc, and centrifuged at $6790 \times g$ for 10 min. The final precipitate was dispersed in 5 mL hexane and stored in a refrigerator at 5 °C before use.

### Preparation of $Cs_{1-x}FA_xPbI_3$ PQDs

The concentrations of the initial $CsPbI_3$ and $FAPbI_3$ PQD solutions were calibrated by the absorption spectra. As $CsPbI_3$ and $FAPbI_3$ PQD solutions have similar concentrations, these two PQDs solutions were mixed in different volume ratios to produce $Cs_xFA_{1-x}PbI_3$ QDs with the desired Cs/FA stoichiometry. The mixtures were sealed and stirred slowly at room temperature for at least 48 h.

### Characterizations of PQDs

UV−Vis spectra were obtained with Varian Cary 100 spectrometer in absorbance mode. NMR measurements were recorded on a Bruker Avance NEO spectrometer operating at a 1H frequency of 400.11 MHz with 32 cycles. The sample temperature was set to 298.2 K. For 1D $^1H$ measurements, 64k data points were sampled with the spectral width set to 20 ppm and a relaxation delay of 1 s. As for all $Cs_xFA_{1-x}PbI_3$ PQD samples, the PQD solutions with same concentrations were dried up completely using $N_2$ flowing, and then 1 mL of benzene-d6 was added to the dried PQDs. After shaking for 5 min, 500 μL of such PQD suspension solution was added to a standard 5 mm NMR tube for measurements. Ex situ and in situ XRD patterns were obtained with Rigaku miniFlex II and Rigaku Smart Lab (Cu Kα radiation, $\lambda = 1.5418$ Å), respectively. In detail, in situ XRD analyses were performed with a heating speed of 20 °C/min and a duration time of 210 s from 30 °C to 550 °C under flowing argon atmosphere. Ex situ XRD measurements were conducted at room temperature in ambient air. TG measurements were carried out on Mettler-Toledo instrument in an argon atmosphere with a heating speed of 10 °C/min. The $Cs_xFA_{1-x}PbI_3$ PQD samples for XRD and TG measurements are prepared by dropping the concentrated PQD solutions onto the center of a Pt substrate and a calcined $Al_2O_3$ crucible, respectively. Steady-state photoluminescence emissions were determined by Techcomp FL970 with a monochromatized Xe lamp as the source excitation when the QD samples were excited at 450 nm. The absolute PLQYs of the PQDs solution were measured at room temperature on Edinburgh FLS1000 equipped with an integrating sphere when the QD samples were excited at 450 nm. Temperature-dependent PL measurements were performed by placing the sample in a vacuum liquid nitrogen thermostat (Cryo-77, Oriental Koji) with a temperature controller, and the temperature is well-controlled from 80 to 390 K. A 405 nm continuous wave (CW) laser (UV−FN−532−500 mW, CNI) was used as the excitation source, and a spectrometer (HR4000CG−UV−NIR, Ocean Optics) was used to collect the spectra. Transmission electron microscope (TEM) and scanning electron microscope (SEM) images were recorded with a JEM-2800 and JSM-7800F, respectively. The PQD samples for SEM measurements were prepared via spin-coating PQD solution on substrate (FTO glass) to obtain flat and smooth PQD thin film, and then heated to the target temperature using a hot plate with a duration time of 180 s. All PQD samples are prepared without any further treatment or additional functionalization, showing their intrinsic properties.

### Computational details for $Cs_{1-x}FA_xPbI_3$ PQDs

All spin-polarized density-functional theory computations were performed using the Vienna ab initio simulation package (VASP5.4.4). The general gradient approximation (GGA) with Perdew-Burke-Ernzerhof (PBE) method was utilized for electron-electron exchange and correlation interactions. Besides, DFT-D3 method with Becke-Johnson damping function was used for Van der Waals interaction while standard PAW potentials were used to describe electron-ion interactions with valence configurations of $2s^22p^3$ for N, $5s^25p^5$ for I, $5s^25p^66s^1$ for Cs, $2s^22p^2$ for C, $6s^26p^2$ for Pb and $1s^1$ for H. A plane-wave basis set was employed to expand the smooth part of the wave functions with a cutoff kinetic energy of 420 eV. To study the mechanistic chemistry of surface reactions, the surface was modelled with a slab model. A sufficiently large vacuum region of 25 Å was used to ensure the periodic images were well separated. The Brillouin-zone integrations were conducted using Monkhorst-Pack grids of special points with a separation of 0.04 $Å^{-1}$. The ligand molecules were calculated in a $30 \times 10 \times 10$ $Å^3$ box. The Brillouin-zone integrations were performed using the Gamma-point-only grid. The convergence criterion for the electronic self-consistent loop was set to $10^{-5}$ eV and the atomic structures were optimized until the residual forces were below 0.03 eV $Å^{-1}$.

## Data availability

The data generated in this study are provided in the Supplementary Information/Source Data file. Data that support the plots within this work are available from the corresponding author upon request. Source data are provided with this paper.

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

## Acknowledgements

This work was supported by the National Natural Science Foundation of China: 22279066 (G.R.L) and 52102266 (Q.Z.), and the China Post-doctoral Science Foundation: 2020M680861 (Q.Z.). This work was done in part at the National Renewable Energy Laboratory, operated by Alliance for Sustainable Energy, LLC, for the U.S. Department of Energy (DOE) under Contract No. DE-AC36-08GO28308 (J.M.L). This work (the initial in situ XRD experiment) was supported by the Center for Hybrid Organic Inorganic Seminconductors for Energy (CHOISE), an Energy Frontier Research Center funded by the Office of Basic Energy Sciences, Office of Science, within the US DOE. The authors wish to thank Pan Zhang at the NKU School of Materials Science and Engineering for high-temperature X-ray diffraction testing. The views expressed in the article do not necessarily represent the views of the DOE or the U.S. Government.

## Author contributions

S.W. Q.Z. and G.L. conceived the project. S.W. and S.L. synthesized the quantum dots and prepared samples for measurements. J.M.L., A.H., and Q.Z. performed the initial in situ XRD measurements. Y.W. performed the SEM measurements. S.W. and Q.Z. performed in situ PL measurements and analysed the data with Y.Z. and X.C. G.L. and Q.Z. supervised the entire project. Q.Z. and S.W. wrote the first draft of the paper. All authors discussed the results and contributed to the writing of the paper.

## Competing interests

The authors declare no competing interests.
