## [Peer Review File · Nature Communications]

Thermal tolerance of perovskite quantum dots dependent on A-site cation and surface ligandREVIEWER COMMENTS

Reviewer #1 (Remarks to the Author):

In this work, Wang and co-workers establish the relation between the chemical composition and the surface ligand binding on thermal behaviors of $Cs_xFA_{1-x}PbI_3$ PQDs across the entire compositional range by systematically performing various in-situ measurements. As results, they think that FA-rich PQDs decompose into PbI_2 directly while Cs-rich PQDs first convert from black orthorhombic γ -phase to yellow orthorhombic δ -phase with increasing temperature. They proposed that the difference of degradation behavior can be attributed to not only the variation of chemical composition, but also the ligand binding energy. Finally, the photogenerated excitons in FA-rich PQDs are easy to be dissociated by LO phonon scattering than that in Cs-rich PQDs. This paper is potentially publishable eventually because of the interesting topic and huge amount of work by authors, however the lack of clarity on the mechanism and theoretical depth need to be more strengthened, prevent me from recommending publish at this time.

1. In page 3, line 74, the ref 26 not mention that “ $CsPbI_3$ PQDs exhibit better photo- and thermal stability as compared to $CsPbBr_3$ ”, they only report that “ $CsPbI_3$ -MeOAc nanocrystals are the most prone to degradation under heating. $CsPbI_3$ nanocrystals exhibit the most stable photophysical states by single crystal luminescence analysis, but eventually degrade under light and readily degrade under heating.” Authors should modify this expression.

2. In Fig. S2, authors claim that the all PQDs retain the original size and cubic shape after alloying, but I think that the size of PQDs gradually become larger as the increase of the amount of the Cs. Also, the morphology are different, the $FAPbI_3$ exhibit cubic cube but the $CsPbI_3$ PQDs seems like the nanosheets. So authors should check the expression. I suggest that author adding the Histograms of size distribution. Also, Boote et al. notice that the photostability of large $CsPbI_3$ PQDs is higher than that of small ones. So authors should analysis if the size of PDQs could affect the thermal degradation process in this work because of the obviously different between the Cs_xFAPbI_3 .

3. As we all known, for $CsPbI_3$, yellow δ -phase is low temperature phase and black phase is high temperature phase, but in XRD pattern, exhibit the phase transition from black γ -phase (~ 30 – 130 °C) to yellow δ -phase (~ 130 – 300 °C), is seems contradictory, author should Further analysis the mechanism of phase transition.

4. In page 8 line 207, author claim that “It is well known that the thermal stability of all inorganic perovskite materials towards retaining chemical composition is higher than that of organic and organic-inorganic hybrid halide perovskites, but which stands in contrast to the case in their nanocrystal counterparts as discussed above”. But I think all inorganic perovskite materials exhibit the better thermal stability towards retaining chemical composition because the FA^+ have evaporated during the heating process leaving behind only PbI_2 , and $CsPbI_3$ only occur the phase transition in the same condition rather than change the composition. How do you think for this expression.

Reviewer #2 (Remarks to the Author):

Summary:

This study focuses on the temperature-dependent behaviors of Cs_xFA_{1-x}PbI₃ perovskite quantum dots across different A-site composition range with extensive in-depth analysis from both experimental measurements and theoretical calculations. The first part of the work reveals different degradation behaviors of Cs_xFA_{1-x}PbI₃ PQDs and links it to the impact of ligand binding energies, supported by XRD, SEM, TGA analysis at elevated temperature and DFT calculations. The second part of the work discusses temperature dependent optical performances of Cs_xFA_{1-x}PbI₃ PQDs, providing initial indication of FA-rich PQDs more favorable for photovoltaic applications.

This work did a comprehensive analysis on thermal properties of Cs_xFA_{1-x}PbI₃ perovskite PQDs, to the best of reviewer's knowledge, no similar work has been done for organic-inorganic-PbI₃ perovskite PQDs.

However, the reviewer still has several questions/suggestions regarding the results discussions. The reviewer thinks it's publishable considering the quality and novelty of the manuscript if authors can make clarifications on below comments:

1. This work applied different measurement techniques with increasing temp heating process, How the heating speed or heating duration at different temp were conducted or controlled? Were they remained same for each sample (different ratio of Cs_xFA_{1-x}PbI₃ PQDs) and for each technique (XRD, SEM, TGA etc) for a fair comparison?

2. In the manuscript, authors claimed 'The black-phase perovskite is completely decomposed into PbI₂ at 350 °C. Then the PbI₂ starts to melt and evaporate under the argon flowing. Above 400 °C, only the peaks from the Pt substrate can be observed at 31.3°, 35.8°, and 39.5°.' in in situ XRD results (Fig 1). However, the TGA results reviewed PbI₂ just started to evaporate until 470 °C. Please explain the discrepancy.

3. Reviewer observes negligible yellow nanorods and less hexagonal PbI₂ in Cs_{0.5}FA_{0.5}PbI₃ compared to pure FAPbI₃ from SEM results and XRD results also showed similar. It seems Cs_{0.5}FA_{0.5}PbI₃ showed best thermal stability among all candidates. Please explain the statement of 'FA-rich PQDs have better thermal stability of Cs-rich.'

4. The author claimed 'the transformation to δ -phase nanorods is first noticeable in CsPbI₃ PQDs at 130 °C, and these ~8 μ m nanorods gradually turn into ~2 μ m perovskite bulks as the temperature is increased to 280 °C.' However, in the XRD results in Fig.1, at 280 °C, there is no signals for PVK bulks yet and was mostly dominated by yellow phase and PbI₂. Please clarify the discrepancy.

5. Since authors suggest FA-rich PQDs more favorable for photovoltaic applications, if any PV devices have or plan to be fabricated to support the statement?

6. Suggestions for minor revisions:

1) Dark brown color for PVK bulk label in Fig 1 is hard to be noticeable.

2) In Page 12, 'As illustrated in Fig. 3C, the calculation results of formation energies strictly for the perovskite phase with corner shared octahedra also explain why FA-rich PQDs have the higher thermal stability than Cs-rich PQDs.' It should be Fig.4C instead of Fig. 3C.

Reviewer #3 (Remarks to the Author):

Shuo Wang et al., reported the structural stabilities of Cs_xFA_{1-x}PbI₃ perovskite quantum dots in the aspect of chemical composition and binding energy of surface ligands. This work is well organized and results are interesting and important. The reviewer read this manuscript with interesting. It worth to be published nature communication after revision.

1. Cs_xFA_{1-x}PbI₃ QDs showed the different PL emissions according to FA/Cs ratio. It means that the each of PQDs have different size and structure, it might affect structural stabilities. Therefore, it might be need to prepare PQDs with similar PL emissions for investigating chemical composition and binding energy of surface ligand effects on stability. The author explains this.

2. Add the size distribution of all CsFAPbI₃, Cs_xFA_{1-x}PbI₃ and FAPbI₃ PQDs.

3. Add PLQY of all CsFAPbI₃, Cs_xFA_{1-x}PbI₃ and FAPbI₃PQDs.

4. Based on SI results, author mentioned that all CsFAPbI₃, Cs_xFA_{1-x}PbI₃ and FAPbI₃ PQDs have a cubic structure. However, peak positions of (002) are different. It means that the crystal structures of PQDs are different. Explain it.

5. Page 4, 111-112, The author mentioned full A-site tuning of Cs_xFA_{1-x}PbI₃ PQDs is well realized based on XRD and morphology results. The reviewer cannot agree it. Author must conduct in-depth XPS and supply chemical composition or carbon/Cs ratio from surface to core.

6. Page 5, 132-133, "Which indicates that not all the FAPbI₃ PQDs decompose into PbI₂ at the same time and part of QDs undergo grain growth and/or merging during this process." The reviewer can not understand scientific meaning of above sentence. Explain it in detail

7. Page 6, 178-179, "As a result, it can be seen the linear thermal expansion coefficient increases with increasing the Cs content of PQDs." Why linear thermal expansion of perovskite QDs increase as Cs content increase. Explain it and add references.

8. The author mentioned that thermal stabilities of all inorganic perovskite materials are higher than those of organic and organic-inorganic hybrid halide perovskites. However, in this research, it is revealed that FAPbI₃ PQDs showed the higher stabilities than those of CsFAPbI₃, Cs_xFA_{1-x}PbI₃ PQDs. Is this case only happening I₃ perovskite and/or FA perovskite QDs? Is this case also happening when Br or other A cation is used?

9. No Figure 3c in Text. And no explanation of some SI figures (Fof example S3-S6). Check it.

10. Page 12, 298-300, "As illustrated in Fig. 3C, the calculation results of formation energies strictly for the perovskite phase with corner shared octahedra also explain why FA-rich PQDs have the higher thermal stability than Cs-rich PQDs." The reviewer can not understand scientific meaning of above sentence. Explain it in detail

11. Page 15,353, Not Figure 4C. Check it.

Reviewer #4 (Remarks to the Author):

The manuscript examines the thermal degradation mechanism of the CsPbI₃, FAPbI₃ and mixed A-site composition of them. The results show that the thermal degradation processes are different for different compositions of PQDs. While the FA rich PQDs decomposes into PbI₂ at elevated temperatures, Cs rich PQDs has gone into a phase transition before decomposing into PbI₂. In the manuscript XRD measurements are studied to define the phase transitions, TGA measurements are investigated the degradation mechanisms in the presence of organic ligands and temperature dependent PL spectroscopy reveals the exciton-phonon interactions at elevated temperatures. Although the study is significant in understanding the fundamentals of thermal stress induced degradation processes in mixed A-site FA_xCs_{1-x}PbI₃ QDs, below points need to be addressed for a complete explanation of some results and discussions. Therefore, I recommend a major revision of this manuscript before publication.

- 1) Degradation mechanisms for the Cs- and FA-rich PQDs are studied in detail in comparison with their pure counterparts. However, I couldn't find enough discussion in the manuscript explaining the degradation mechanism for Cs_{0.5}FA_{0.5}PbI₃.
- 2) It is important to investigate the degradation mechanism of these mixed A-site PQDs. However the devices using these QDs will not going to be working at these elevated temperatures. For practical point of view, I find it useful to include the possible cooling down phase transformation of these PQDs after heated up to the phase transitions temperatures.
- 3) The authors claim in the conclusion that FA-rich PQDs are favorable for PV applications based on the conclusion that photogenerated excitons are dissociating by LO photon scattering. However this claim is based on the fitting at elevated temperatures without considering ionized impurities. Also it is well known in the literature that ionized impurities has significant effect on the PV device performance. Considering the operating temperature range of PV devices, can the authors comment on the validity of this claim?
- 4) In Fig 5a, temperature dependent PL spectra is given. Does the authors observe any significant change in the PL spectra at the phase transition temperatures? Also, it is shown in the figures that PL FWHM narrows at temperatures between 250°C and 300°C in all PQDs except FAPbI₃. Can the authors comment on this behavior?

Response to the Reviewers' Comments

Reviewer #1 (Remarks to the Author):

In this work, Wang and co-workers establish the relation between the chemical composition and the surface ligand binding on thermal behaviors of $\text{Cs}_x\text{FA}_{1-x}\text{PbI}_3$ PQDs across the entire compositional range by systematically performing various in-situ measurements. As results, they think that FA-rich PQDs decompose into PbI_2 directly while Cs-rich PQDs first convert from black orthorhombic γ -phase to yellow orthorhombic δ -phase with increasing temperature. They proposed that the difference of degradation behavior can be attributed to not only the variation of chemical composition, but also the ligand binding energy. Finally, the photogenerated excitons in FA-rich PQDs are easy to be dissociated by LO phonon scattering than that in Cs-rich PQDs. This paper is potentially publishable eventually because of the interesting topic and huge amount of work by authors, however the lack of clarity on the mechanism and theoretical depth need to be more strengthened, prevent me from recommending publish at this time.

Response: Thanks for your positive comments on this work. You can find we have made the discussions clearer and stronger with point-to-point response to your questions and suggestions.

1. In page 3, line 74, the ref 26 not mention that “ CsPbI_3 PQDs exhibit better photo- and thermal stability as compared to CsPbBr_3 ”, they only report that “ CsPbI_3 -MeOAc nanocrystals are the most prone to degradation under heating. CsPbI_3 nanocrystals exhibit the most stable photophysical states by single crystal luminescence analysis, but eventually degrade under light and readily degrade under heating.” Authors should modify this expression.

Response: Thanks for the comment. The expression has been changed in the revised manuscript as “Recent work by Boote *et al.* shows with the help of X-ray diffraction (XRD) and thermogravimetric analysis (TGA) that CsPbI_3 PQDs exhibit better photostability but worse thermal stability as compared to CsPbBr_3 , and also exhibit different degradation behavior from CsPbCl_3 PQDs.”

2. In Fig. S2, authors claim that the all PQDs retain the original size and cubic shape after alloying, but I think that the size of PQDs gradually become larger as the increase of the amount of the Cs. Also, the morphology is different, the FAPbI_3 exhibit cubic cube but the CsPbI_3 PQDs seems like the nanosheets. So, authors should check the expression. I suggest that author adding the Histograms of size distribution. Also, Boote *et al.* notice that the photostability of large CsPbI_3 PQDs is higher than that of small ones. So, authors should analysis if the size of PDQs could affect the thermal degradation process in this work because of the obviously different between the CsFAPbI_3 .

Response: Thank you so much for the suggestions. We re-performed TEM

observations for all kinds of PQDs, and presents the images and size distributions in **Fig. S3**. The mean sizes for CsPbI₃, Cs_{0.75}FA_{0.25}PbI₃, Cs_{0.5}FA_{0.5}PbI₃, Cs_{0.25}FA_{0.75}PbI₃, and FAPbI₃ PQDs are 10.48 nm, 10.90 nm, 11.21 nm, 11.18 nm, and 11.35 nm, respectively. The FA-rich PQDs exhibit slightly larger crystal size (< 1 nm, or negligible) than Cs-rich PQDs. However, in the paper by Boote et al., the size for large CsPbI₃ PQDs (~34 nm) is more than two times larger than that for small CsPbI₃ PQDs (~14 nm), which results in the difference of degradation behaviour. Therefore, although we do agree that the size of PQDs affects the thermal degradation process, we believe that the difference in degradation behaviour between different PQDs, presented in our work, is mainly attributed to chemical composition and ligand binding, rather than the size. With regards to the nanosheet issue, there will be a strong PL emission peak at ~600 nm if the CsPbI₃ nanocrystals are in the shape of nanosheet, as shown in the following **Figure R1** by Lv et al. (DOI: 10.1039/C6NR03428D). In addition, CsPbI₃ nanosheets are generally synthesized via a hot-injection method below 140 °C, while CsPbI₃ PQDs reported here are synthesized at 180 °C following the previous reports. (*Science* **354**, 92–95 (2016); *ACS Nano* **12**, 10327–10337 (2018); *ACS Energy Lett.* **4**, 1954–1960 (2019); *Nat. Commun.* **10**, 2842 (2019)) Following the suggestion, the manuscript is modified as “The morphologies and size distributions of Cs_xFA_{1-x}PbI₃ PQDs are displayed in Fig. S3, showing all PQDs retain the original cubic shape with a negligible mean size difference (< 1 nm) after alloying.”

Figure R1. PL spectra of CsPbX₃ nanosheets with different halide compositions. (DOI: 10.1039/C6NR03428D)

Fig. S3. (A–E) TEM images (left panel) and size distribution (right panel) of CsPbI₃ (A), Cs_{0.75}FA_{0.25}PbI₃ (B), Cs_{0.5}FA_{0.5}PbI₃ (C), Cs_{0.25}FA_{0.75}PbI₃ (D), and FAPbI₃ PQDs (E).

3. As we all known, for CsPbI₃, yellow δ -phase is low temperature phase and black phase is high temperature phase, but in XRD pattern, exhibit the phase transition from black γ -phase ($\sim 30\text{--}130\text{ }^\circ\text{C}$) to yellow δ -phase ($\sim 130\text{--}300\text{ }^\circ\text{C}$), is seems contradictory, author should further analysis the mechanism of phase transition.

Response: Thanks for the comment. For CsPbI₃, non-perovskite δ -phase with wide band gap (E_g : ~ 2.82 eV, yellow) is thermodynamically more stable than perovskite α -phase, β -phase, and γ -phase (The difference between these three phases is only on the degree of bond tilting, E_g : ~ 1.73 eV, black) below ~ 320 °C in general. So, black-phase CsPbI₃ undergo immediate transformation to the yellow phase when exposed to ambient conditions. However, Luther et al. demonstrated that surface strain imparted by the organic ligands on the QD surfaces can be used to stabilize black-phase CsPbI₃ at room temperature, far below the phase transition temperature for thin film or bulk materials, which also enables the high-performance solar cells (*Science* **354**, 92–95 (2016)). Since then, black-phase CsPbI₃ PQDs are widely used in devices, particularly in photovoltaic devices. Furthermore, in another work, we reveal that the large CsPbI₃ PQDs exhibit γ -phase, rather than previously reported α -phase, due to the negative surface energy (*ACS Energy Lett.* **5**, 238–247 (2020)). Therefore, CsPbI₃ PQDs prepared following the method in our previous reports shows black γ -phase at room temperature. When heating CsPbI₃ PQDs up to ~ 130 °C, the surface ligands start to evaporate exposing the QDs directly to the environment and lifting the surface strain, and thus losing stability of the black-phase CsPbI₃. Then, the black γ -phase of CsPbI₃ PQDs is changed to non-perovskite δ -phase that is thermodynamically more stable in the range of ~ 130 – 300 °C. The modifications are shown in the revised manuscript as “For CsPbI₃ PQDs, the black phase can be stabilized at room temperature by bonding with surface ligands. When heating up to ~ 130 °C, the surface ligands are evaporated, and the black γ -phase of CsPbI₃ PQDs is changed to the orthorhombic yellow phase which is more thermodynamically stable in the range of ~ 130 – 300 °C, as described above.”

4. In page 8 line 207, author claim that “It is well known that the thermal stability of all inorganic perovskite materials towards retaining chemical composition is higher than that of organic and organic-inorganic hybrid halide perovskites, but which stands in contrast to the case in their nanocrystal counterparts as discussed above”. But I think all inorganic perovskite materials exhibit the better thermal stability towards retaining chemical composition because the FA⁺ have evaporated during the heating process leaving behind only PbI₂, and CsPbI₃ only occur the phase transition in the same condition rather than change the composition. How do you think for this expression.

Response: Thanks. We agree that “all inorganic perovskite materials exhibit the better thermal stability towards retaining chemical composition because of the evaporation of FA”. The crystal phase of PQDs used as light harvesting materials in device applications is usually black perovskite phase. In another word, when we talk about stability here, we consider the stability of PQD materials in the black perovskite phase. Therefore, to accurately describe it, we modify the paper as “It is well known that the thermal stability of all inorganic perovskite materials towards retaining the desirable crystal phase (black phase) for device applications as light harvesting materials, is higher than that of organic and organic-inorganic hybrid halide perovskites,^{30,31} but which stands in contrast to the case in their nanocrystal counterparts as discussed above.”

Reviewer #2 (Remarks to the Author):

Summary: This study focuses on the temperature-dependent behaviors of $Cs_xFA_{1-x}PbI_3$ perovskite quantum dots across different A-site composition range with extensive in-depth analysis from both experimental measurements and theoretical calculations. The first part of the work reveals different degradation behaviors of $Cs_xFA_{1-x}PbI_3$ PQDs and links it to the impact of ligand binding energies, supported by XRD, SEM, TGA analysis at elevated temperature and DFT calculations. The second part of the work discusses temperature dependent optical performances of $Cs_xFA_{1-x}PbI_3$ PQDs, providing initial indication of FA-rich PQDs more favourable for photovoltaic applications. This work did a comprehensive analysis on thermal properties of $Cs_xFA_{1-x}PbI_3$ perovskite PQDs, to the best of reviewer's knowledge, no similar work has been done for organic-inorganic- PbI_3 perovskite PQDs. However, the reviewer still has several questions/suggestions regarding the results discussions. The reviewer thinks it's publishable considering the quality and novelty of the manuscript if authors can make clarifications on below comments:

Response: Thank you very much for approving the quality and novelty of this work. We have improved it following with your questions and suggestions.

1. This work applied different measurement techniques with increasing temp heating process, how the heating speed or heating duration at different temp were conducted or controlled? Were they remained same for each sample (different ratio of $Cs_xFA_{1-x}PbI_3$ PQDs) and for each technique (XRD, SEM, TGA etc) for a fair comparison?

Response: Thank you for the interesting questions. The measurement conditions in all in situ characterizations are remained same for each sample, which is an essential precondition for a fair comparison between different PQDs in one characterization. Although we think that achieving a fair comparison between different measurement techniques is ideal for this work, it is difficult to make it happen in reality, due to the different levels of sensitivity, detectivity, software programs and accuracy rates for different characterization equipment, as well as the different amounts of PQD samples required for different measurement techniques. For example, the PQD samples for SEM measurements are prepared via spin-coating diluted QD solution on substrate (FTO glass) to obtain flat and smooth QD thin film, whereas the samples for XRD and TGA measurements are prepared via dropping concentrated QD solution into the particular holder of the instrument to guarantee that there are enough amounts of QDs for relatively long-time measurements. In addition, there is no duration time that can be set in TG heating program, which results in large differences in heating process between XRD and TGA. Therefore, we would not suggest to compare the temperatures obtained in different kinds of measurements though we have tried our best to keep some of the measurement condition same. In this work, it can be found that the main trends and conclusions drawn from different measurements are mostly consistent. Following your suggestion, we reperformed in situ XRD on pure $CsPbI_3$, $Cs_{0.5}FA_{0.5}PbI_3$ and pure $FAPbI_3$ PQDs to check the accuracy and repeatability, as demonstrated in **Figure R2**. The degradation process shows a little difference in temperature, but quite same trends. The experimental section is also revised as “In detail, *in situ* XRD analyses were

performed with a heating speed of 20 °C/min and a duration time of 210s from 30 °C to 550 °C under flowing argon atmosphere, and *ex situ* XRD were conducted at room temperature in ambient air. TGA were carried out on Mettler-Toledo instrument in an argon atmosphere with a heating speed of 10 °C/min. The $Cs_xFA_{1-x}PbI_3$ PQD samples for XRD and TG measurements are prepared by dropping the concentrated PQD solutions onto the center of a Pt substrate and a calcined Al_2O_3 crucible, respectively. Transmission electron microscope (TEM) and scanning electron microscope (SEM) images were recorded with a JEM-2800 and JSM-7800F, respectively. The PQD samples for SEM measurements are prepared via spin-coating diluted PQD solution on substrate (FTO glass) to obtain flat and smooth QD thin film, and then heated to the target temperature using a hot plate with a duration time of 180s.”

Figure R2. In situ XRD for pure $CsPbI_3$, $Cs_{0.5}FA_{0.5}PbI_3$ and pure $FAPbI_3$ PQDs re-performed for checking the accuracy of the measurement.

2. In the manuscript, authors claimed ‘The black-phase perovskite is completely decomposed into PbI_2 at 350 °C. Then the PbI_2 starts to melt and evaporate under the argon flowing. Above 400 °C, only the peaks from the Pt substrate can be observed at 31.3°, 35.8°, and 39.5°.’ in in situ XRD results (Fig 1). However, the TGA results reviewed PbI_2 just started to evaporate until 470 °C. Please explain the discrepancy.

Response: Thank you for this comment. It must be admitted that the temperature points obtained in different measurements (such as for phase transition or emerging, melting and evaporation of PbI_2) are not exactly same. There are some possible reasons for this discrepancy: (1) The amount and the exposed surface area (or heating area) of PQD samples are different. To give a simple example, it needs less time and lower heating temperature to completely evaporate a small amount of a substance with a large surface area than that for a large amount with a small exposed surface area. As shown in the following **Fig. S15**, the amount of PQD samples: SEM \ll XRD $<$ TGA; the exposed area of PQD samples: SEM \approx XRD \gg TGA. Therefore, we can expect that the temperature point for the change of PQD samples: TGA $>$ XRD $>$ SEM. (2) As mentioned in the response for the Comment 1, there is no duration time that can be set in TG heating program, which results in large differences in heating process between XRD and TGA. It suggests that the heating time during the TGA measurement is shorter than that in XRD measurement. (3) The instrument parameters for XRD and TGA measurements (such as accuracy rate, sensitivity, program system, etc.) are different. What we can guarantee is that the measurement conditions for each sample in one kind of characterization are remained same, which can enable a fair comparison between different PQDs. Following your suggestions, we modify the manuscript as “It should be noted that the discrepancy in the temperature points for the changes of PQD samples (such as the phase transitions, the emerging, melting and evaporation of PbI_2) between different measurement techniques may be ascribed to the differences in the heating process, the instrument parameters (accuracy rate, sensitivity and program system, etc.), as well as the amount and the exposed surface area of the PQD samples used in characterizations (Fig. S15).”

Fig. S15. (A-C) Photographs of PQR samples used in SEM (A), in situ XRD (B), and TGA (C) measurements with their corresponding holders or substrates.

3. Reviewer observes negligible yellow nanorods and less hexagonal PbI_2 in $\text{Cs}_{0.5}\text{FA}_{0.5}\text{PbI}_3$ compared to pure FAPbI_3 from SEM results and XRD results also showed similar. It seems $\text{Cs}_{0.5}\text{FA}_{0.5}\text{PbI}_3$ showed best thermal stability among all candidates. Please explain the statement of ‘FA-rich PQRs have better thermal stability of Cs-rich.’

Response: Thanks for the suggestion while we are sorry for this misunderstanding. The XRD and SEM presented here are not to compare the amount of degradation products, but rather to demonstrate the process of phase transition and degradation of different PQRs. The amount of degradation products in SEM images and the intensity of XRD results without any normalization depend on how many PQRs are in the scanning area

during characterization, as shown in **Fig. S15** (they are not perfectly uniform, and can be chosen purposely). Moreover, we must also admit that, based on the SEM and XRD results, it is difficult to identify which PQD have apparently higher thermal stability than others. The aim of this work is to attempt to answer why and how CsPbI₃ (or Cs-rich) PQDs show similar or even slightly lower stability than FAPbI₃ (or FA-rich) PQDs which is in contrast with what we believe: all inorganic QDs possess much higher thermal stability than organic ones. To avoid this misunderstanding, we have added it in the revised manuscript as “The amount of degradation products in SEM images and the intensity of XRD results without any normalization depend on how many PQDs are in the scanning area during characterization, as shown in Fig. S15.”

4. The author claimed ‘the transformation to δ -phase nanorods is first noticeable in CsPbI₃ PQDs at 130 °C, and these ~8 μ m nanorods gradually turn into ~2 μ m perovskite bulks as the temperature is increased to 280 °C.’ However, in the XRD results in Fig. 1, at 280 °C, there is no signals for PVK bulks yet and was mostly dominated by yellow phase and PbI₂. Please clarify the discrepancy.

Response: Thanks for pointing this out. Based on the analysis of in situ XRD results, the degradation of CsPbI₃ PQDs under thermal stress could be mainly attributed to the phase transition from black γ -phase (~30–130 °C) to yellow δ -phase (~130–300 °C) and then finally to black α -phase (~300–450 °C) of perovskite, along with a tiny decomposition into PbI₂ since only a few characteristic diffraction peaks of PbI₂ can be observed during the whole period of the heating process. However, we do not observe noticeable hexagonal PbI₂ during degradation in SEM for CsPbI₃ PQDs which can be identified in XRD patterns. Furthermore, the temperature point obtained from SEM measurements are usually lower than that from XRD results. These discrepancies are possibly attributed to the differences in the amount of PQD samples and the heating program process used in measurements, as described in the response of Comment 1 and 2. The PQD samples for XRD measurements are prepared by dropping the concentrated PQD solutions onto the center of a Pt substrate, having a much more amount of QDs than the SEM sample which is prepared via spin-coating diluted PQD solution on FTO glass. The large amount of QDs in XRD measurement is required for high signal intensity, while the small amount of QDs (thin film) in SEM measurement is beneficial to get high resolution images due to high conductivity. Therefore, this description has been revised in the updated version as “In addition, we must note that here the amount of degradation products displayed in SEM images and the intensities of XRD peaks without any normalization are closely related to how many PQDs are in the scanning area during characterization (Fig. S15), which could not represent the stability of PQDs since the samples are not perfectly uniform and the area can be chosen purposely. Moreover, we must also admit that, based on the SEM and XRD results, it is difficult to identify which PQDs have apparently higher thermal stability than the others. In other words, Cs-rich PQDs exhibit similar or even slightly poorer thermal stability of initial black perovskite phase than FA-rich PQDs, owing to the black-to-yellow phase transition at relatively low temperature.”

5. Since authors suggest FA-rich PQDs more favorable for photovoltaic applications, if any PV devices have or plan to be fabricated to support the statement?

Response: Thanks, and we appreciate this comment and suggestion. As we all know, high performance and high stability of devices require large optimization work, which is associated with the film thickness, device architecture, ligand manipulation, choosing of hole and electron transport materials, etc. Currently, the optimization work on solar cells based on different PQDs is being carried out in our lab, and we hope these device research work can be published in the near future. Although the device research is not ready-to-published in the current stage, which is also out of the scope of this work, many related works by other researchers have been published recently. In **Figure R3**, we present some of excellent works on the stability of solar cells based on FAPbI₃ and CsPbI₃ PQDs in 2022. It seems that FAPbI₃ PQD solar cells show better stability than CsPbI₃ PQD devices, which is in agreement with our results in this work.

Figure R3. Reports on the stability test of solar cells based on FAPbI₃ and CsPbI₃ PQDs in 2022

6. Suggestions for minor revisions:

1) Dark brown color for PVK bulk label in Fig 1 is hard to be noticeable.

Response: Thanks for suggestion. The Fig.1 is modified as:

Fig. 1. *In situ* XRD analyses of Cs_xFA_{1-x}PbI₃ PQDs. (A) *In situ* XRD patterns collected from 30 °C to 500 °C under argon flowing for FAPbI₃, Cs_{0.25}FA_{0.75}PbI₃, Cs_{0.5}FA_{0.5}PbI₃, Cs_{0.75}FA_{0.25}PbI₃, and CsPbI₃ PQDs. (B) Scheme for the dominant reflections in the thermal degradation of Cs_xFA_{1-x}PbI₃ PQDs that is drawn from the vertical line cuts of the 2-dimensional *in situ* XRD patterns taken at the 2θ angles shown in Table S1 for perovskite in QD form (PVK-QDs), perovskite in bulk form (PVK-bulk films), orthorhombic non-perovskite (non-PVK yellow phase), and PbI₂. The peak of substrate is marked with asterisks.

2) In Page 12, ‘As illustrated in Fig. 3C, the calculation results of formation energies strictly for the perovskite phase with corner shared octahedra also explain why FA-rich PQDs have the higher thermal stability than Cs-rich PQDs.’ It should be Fig.4C instead of Fig. 3C.

Response: Thanks. The sentence is changed as “As illustrated in Fig. 4C, FA-rich PQDs exhibit more negative values of formation energy than Cs-rich PQDs, suggesting that the FA-rich PQDs having the perovskite phase with corner shared octahedra is more thermodynamically favorable at room temperature.”

Reviewer #3 (Remarks to the Author):

Shuo Wang et al., reported the structural stabilities of $\text{Cs}_x\text{FA}_{1-x}\text{PbI}_3$ perovskite quantum dots in the aspect of chemical composition and binding energy of surface ligands. This work is well organized and results are interesting and important. The reviewer read this manuscript with interesting. It worth to be published nature communication after revision.

Response: Thank you very much for so positive comments.

1. $\text{Cs}_x\text{FA}_{1-x}\text{PbI}_3$ QDs showed the different PL emissions according to FA/Cs ratio. It means that the each of PQDs have different size and structure, it might affect structural stabilities. Therefore, it might be need to prepare PQDs with similar PL emissions for investigating chemical composition and binding energy of surface ligand effects on stability. The author explains this.

Response: Thank you for your insightful comment. Firstly, based on your suggestion, we show the PLQY and TRPL of different PQDs in **Fig. S1** in the revised manuscript as “FA-rich PQDs (or pure FAPbI_3 PQDs) exhibit higher PLQY and longer exciton lifetime than Cs-rich PQDs (or pure CsPbI_3 PQDs) as demonstrated in Fig. S1C and S1D.” We absolutely agree that the PL emission in a particular QD composition is associated with crystal size and structure, for example, smaller sized CsPbI_3 PQDs show slightly higher energy PL emission than that of larger size CsPbI_3 PQDs (*ACS Energy Lett.* **5**, 238–247 (2020)). But here, the PL emission energy is different because of variation of A-site composition (*ACS Nano* **12**, 10327–10337 (2018)), and doesn't necessarily mean that the PL emission difference is due to variation of size, shape and structure. To further confirm this, we have added clear TEM images and size distributions into **Fig. S3**, and the manuscript is revised as “The morphologies and size distributions of $\text{Cs}_x\text{FA}_{1-x}\text{PbI}_3$ PQDs are displayed in Fig. S3, showing all PQDs retain the original cubic shape with a negligible mean size difference (< 1 nm) after alloying.” Furthermore, it must be considered that the PL spectra of PQDs also strongly depends on the trap density, interdot spacing, and so on. In this work, all PQDs are prepared by the widely used methods (hot-injection for pure PQDs and anion exchange for the alloys), showing their intrinsic properties, without any further treatment or additional functionalization. However, preparing different PQDs but with similar PL emissions has to need further ligand manipulation, defect passivation or/and other treatments, which is not our focus here. To avoid this misunderstanding, we add a sentence in the revised paper as “All PQD samples are prepared without any further treatment or additional functionalization, showing their intrinsic properties.” For example, well-passivated CsPbI_3 PQDs may have much better stability than intrinsic FAPbI_3 PQDs. Actually, it is consistent with our understanding that the stability of PQDs not only relies on the chemical compositions, but also the ligand binding that influences surface defect passivation, lattice tilting, ligand or ion exchange efficiency etc.

Fig. S1. (A–C) UV-Vis (A), PL spectra (B), PLQY (C), TRPL (D), and XRD patterns (E) of Cs_xFA_{1-x}PbI₃ QDs with different compositions.

Fig. S3. (A–E) TEM images (left panel) and size distribution (right panel) of CsPbI₃ (A), Cs_{0.75}FA_{0.25}PbI₃ (B), Cs_{0.5}FA_{0.5}PbI₃ (C), Cs_{0.25}FA_{0.75}PbI₃ (D), and FAPbI₃ PQDs (E).

2. Add the size distribution of all CsFAPbI₃, Cs_xFA_{1-x}PbI₃ and FAPbI₃ PQDs.

Response: Following your suggestion, the manuscript is revised as “The morphologies

and size distributions of $\text{Cs}_x\text{FA}_{1-x}\text{PbI}_3$ PQDs are displayed in Fig. S3, showing all PQDs retain the original cubic shape with a negligible mean size difference (< 1 nm) after alloying.” (Fig. S3 is shown in Response to Comment 1)

3. Add PLQY of all CsFAPbI_3 , $\text{Cs}_x\text{FA}_{1-x}\text{PbI}_3$ and FAPbI_3 PQDs.

Response: Thanks for the suggestion. The PLQY data are added in Fig. S1 in the revised manuscript. (Fig. S1 is shown in Response to Comment 1)

4. Based on SI results, author mentioned that all CsFAPbI_3 , $\text{Cs}_x\text{FA}_{1-x}\text{PbI}_3$ and FAPbI_3 PQDs have a cubic structure. However, peak positions of (002) are different. It means that the crystal structures of PQDs are different. Explain it.

Response: The size of Cs^+ cation (ionic radius of 0.167 nm) is smaller than that of FA^+ cation (ionic radius of 0.205 nm). Thus, CsPbI_3 PQDs exhibit the smaller distance, d , between atomic layers in crystal structure than FAPbI_3 PQDs. According to Bragg’s Law ($n\lambda=2d\sin\theta$), the XRD peaks (2θ) shift to large diffraction angles as increasing the Cs amount in PQDs. It also should be noted that CsPbI_3 in bulk can have more than one perovskite phase with corner-shared octahedra. For example, non-perovskite δ -phase of CsPbI_3 (orthorhombic, Pnma) can be converted into cubic α phase (with undistorted corner shared $[\text{PbI}_6]^{4-}$ octahedra, Pm-3m) upon heating above 360 °C. There can also be two more perovskite phases of CsPbI_3 at temperatures lower than 360°C depending upon the amount of tilting of the $[\text{PbI}_6]^{4-}$ octahedra: the β phase (260°C) and the γ phase (175°C). Since the PQDs were successfully synthesized by hot injection method (*Nano Lett.* **15**, 3692–3696 (2015)), these kinetically stabilized perovskite phases can exist at room temperature in ambient conditions via ligand binding of long-chain organic ligands (such as OA and OAm). In our previous reports (*ACS Energy Lett.* **5**, 238–247 (2020) *ACS Energy Lett.* **5**, 2475–2482 (2020)), we further reveal that the $\text{Cs}_x\text{FA}_{1-x}\text{PbI}_3$ PQDs exhibit the near-cubic perovskite phase with tilting of the $[\text{PbI}_6]^{4-}$ octahedra at room temperature, black γ -phase or β -phase, although it is difficult to identify the exact amount of octahedral tilting in all $\text{Cs}_x\text{FA}_{1-x}\text{PbI}_3$ PQDs and associated space group due to Scherrer broadening of the XRD peaks. For FAPbI_3 PQDs, they are shown to exist in the cubic perovskite phase with space group Pm-3m. Overall, the monotonically linear shift (based on Vegard’s law) of peak position (002) is mainly attributed from the change of A-site cation. Following your suggestion, the description is modified as “The XRD data show that all the prepared $\text{Cs}_x\text{FA}_{1-x}\text{PbI}_3$ PQDs exhibit black perovskite phase (cubic or near-cubic), and the detailed analysis on the tilting of the $[\text{PbI}_6]^{4-}$ octahedra in these PQDs can be found in the previous work. Here, in order to not increase the complexity of further analysis, we assume that Cs-rich PQDs are in γ -phase, and FA-rich as well as $\text{Cs}_{0.5}\text{FA}_{0.5}\text{PbI}_3$ PQDs are in α -phase. Since CsPbI_3 PQDs exhibit a smaller spacing of the crystal layers than FAPbI_3 PQDs, the diffraction (002) peak shifts monotonically between the CsPbI_3 and FAPbI_3 PQD patterns (Fig. S1E) following Bragg’s and Vegard’s laws.”

5. Page 4, 111-112, The author mentioned full A-site tuning of $\text{Cs}_x\text{FA}_{1-x}\text{PbI}_3$ PQDs is well realized based on XRD and morphology results. The reviewer cannot agree it.

Author must conduct in-depth XPS and supply chemical composition or carbon/Cs ratio from surface to core.

Response: Thank you for the pertinent suggestion. In this work, the $\text{Cs}_x\text{FA}_{1-x}\text{PbI}_3$ PQDs with full A-site tuning are synthesized and characterized following the previous works: *Science* **354**, 92–95 (2016), *ACS Nano* **12**, 10327–10337 (2018), *ACS Energy Lett.* **4**, 1954–1960 (2019), and *Nat. Commun.* **10**, 2842 (2019). The details of the post-synthetic alloying procedure can be found in *ACS Nano* **12**, 10327–10337 (2018). The claim of the full A-site tuning is also backed by monotonic variation of optical absorption and PL fluorescence spectroscopy: the absorption onset and the PL emission peak positions can be continuously and monotonically tuned from pure CsPbI_3 to pure FAPbI_3 within the range of ~ 650 – 800 nm. In our previous report (*ACS Nano* **12**, 10327–10337 (2018)), we alloyed these PQDs of different sizes (i.e., small CsPbI_3 PQDs with PL maximum at ~ 660 nm and larger FAPbI_3 NCs with PL maximum at ~ 770 nm), and observed that an asymmetric absorption with a bimodal distribution of particles and an asymmetric PL emission spectrum that can be deconvoluted with two Gaussians with peak maximum at ~ 680 and ~ 728 nm. This indicates that the alloyed PQDs retain their original size, with the final composition being tuned by the relative amounts of Cs^+ to FA^+ ions. Moreover, in regard of structural characterization, all the alloy compositions retain their perovskite structure, and the diffraction peaks shift monotonically between the patterns obtained with pure CsPbI_3 and FAPbI_3 PQDs according to Bragg's and Vegard's laws. Thus, from synthesis, optical and structural characterizations, the results are all in good agreement with previous reports, showing full A-site tuning of $\text{Cs}_x\text{FA}_{1-x}\text{PbI}_3$ PQDs.

Regarding the reviewer's suggestion for in-depth XPS study to determine the carbon/Cs ratio, we think it is not very trivial with laboratory X-ray sources with standard measurement conditions though it is a very relevant suggestion here. Firstly, as we all known, it is hard to probe one PQD (~ 10 nm) from surface to core based on most of laboratory instruments since the region of analysis usually needs to be > 1 μm in size. Secondly, if we understand correctly, the reviewer is asking to carry out depth dependent XPS to determine the average carbon/Cs ratio from the surface of QD to the core of the QD film, which means that we will collect the signals from thousands of or more PQDs in one scanning of XPS spectrum. The XPS result can be strongly influenced by the roughness of PQD film samples, the ordering of PQD arrays (interdot spacing), and the number of organic ligands left in PQD films. As presented in Figure R4, each probing signal covers surface and inter position simultaneously owing to the disorder of PQD arrays with three-dimensional cubic shape. Furthermore, such depth profiling studies using variable energy synchrotron radiation-based photoelectron facilities have been reported for II-VI metal chalcogenide QDs to determine the internal structures (*J. Am. Chem. Soc.* **131**, 470–477 (2009); *J. Phys. Chem. Lett.* **1**, 2149–2153 (2010); *J. Phys. Chem. C* **118**, 15534–15540 (2014)). Angle resolved photoelectron spectroscopy (ARPES) may also give some insight, but the analysis will be far more complicated, given the fact that the organic ligands at the surface also contribute hugely to the C 1s as well as N 1s signals along with the signal from the A-site FA ion. Such study is beyond the scope of this manuscript. It is to be noted that time-of-flight

secondary ion mass spectrometry (TOF-SIMS) is also another useful technique to extract such in-depth elemental profiling, as we have done in our pervious report. (*Nat. Commun.* **10**, 2842 (2019))

Figure R4. Schematic of PQD samples for XPS

However, with our limited capabilities, we have attempted to perform the XPS experiment on surface etched PQDs to get some more insight. We performed Al K Alpha, 1486.6 eV based XPS profiling on $\text{Cs}_{0.5}\text{FA}_{0.5}\text{PbI}_3$ PQDs with the etching energy of 500 eV, the etching rate of around 0.2 nm/s, and two sets of etching time (0-100s with a step of 25s and 0-200s with a step of 50s). The instrument is Thermo SCIENTIFIC ESCALAB 250 Xi using a spot size of 500 μm .

Figure R5. XPS spectra of $\text{Cs}_{0.5}\text{FA}_{0.5}\text{PbI}_3$ PQDs with an etching time of 100s and a step of 25s

Figure R6. XPS spectra of $\text{Cs}_{0.5}\text{FA}_{0.5}\text{PbI}_3$ PQDs with an etching time of 200s and a step of 50s

It can be found in both Figure R5 and R6 that the intensity of Cs signal is increased monotonically as increasing the etching time, which possibly suggests the distribution of Cs is uniform in the QDs. But, as described above, it may only demonstrate the differences between upper and below PQD layer, which is not our focus in this work. Moreover, it should be noted that the C and N signals are influenced by the organic ligands left in PQD films like OA and OAm. Therefore, we can conclude that in-depth XPS is quite complex in this case.

To best appease the reviewer, we have decided to carrier out ^1H NMR measurements for FAPbI_3 , $\text{Cs}_{0.25}\text{FA}_{0.75}\text{PbI}_3$, $\text{Cs}_{0.5}\text{FA}_{0.5}\text{PbI}_3$, $\text{Cs}_{0.75}\text{FA}_{0.25}\text{PbI}_3$, and CsPbI_3 PQDs, showing the information of chemical compositions in Figure S2. It shows that the integral areas of both C-H (H1) and N-H (H2) in FA^+ cation linearly increase as the FA content increases (the value of x in $\text{Cs}_x\text{FA}_{1-x}\text{PbI}_3$ is changed from 1, 0.75, 0.5, 0.25 to 0). The description has been added in the revised manuscript as “The ^1H nuclear magnetic resonance (NMR) spectra consistently demonstrated that the integral areas of both C-H (H1) and N-H (H2) in FA^+ cation linearly increase as the value of x in $\text{Cs}_x\text{FA}_{1-x}\text{PbI}_3$ decreases from 1 to 0 (Fig. S2).”

Fig. S2. (A–B) ^1H NMR spectra (A) and the integral area of C-H (H1) and N-H (H2) in FA^+ cation (B) for $\text{Cs}_x\text{FA}_{1-x}\text{PbI}_3$ PQDs with different compositions.

6. Page 5, 132-133, “Which indicates that not all the FAPbI_3 PQDs decompose into PbI_2 at the same time and part of QDs undergo grain growth and/or merging during this process.” The reviewer cannot understand scientific meaning of above sentence. Explain it in detail

Response: Thanks for your comment. During the degradation process of FAPbI_3 PQDs, when heating up to around 150°C , the diffraction peaks attributed to PbI_2 start appearing, which indicates that the FAPbI_3 PQDs start to decompose into PbI_2 . In the meantime (temperature increases from 150°C to 300°C), the peaks from the black phase perovskite surprisingly become stronger and sharper. According to Scherrer equation, the larger crystal size shows sharper XRD peak. The overall stronger peak intensities suggest relatively higher crystallinity that is usually demonstrated by large crystal. Therefore, in the range of 150°C to 300°C , FAPbI_3 undergo grain growth to large grains, and the decomposition of part of FAPbI_3 occurs at the same time. The sentence has been modified as “When heating up to around 150°C , the diffraction peaks at 25.2° , 29.0° , and 41.2° attributed to (102), (103), and (006) planes of PbI_2 , respectively, start appearing and intensities increase with increasing temperature, which indicates that the FAPbI_3 PQDs start to decompose into PbI_2 . But, in the meantime, the peaks from the black phase perovskite surprisingly become stronger and sharper as the temperature increases from 150°C to 300°C . Thus, it allows us to state that part of FAPbI_3 PQDs which has not been decomposed into PbI_2 undergo grain growth and/or merging in this stage.”

7. Page 6, 178-179, “As a result, it can be seen the linear thermal expansion coefficient increases with increasing the Cs content of PQDs.” Why linear thermal expansion of perovskite QDs increase as Cs content increase. Explain it and add references.

Response: Thanks a lot for this valuable suggestion. To our best knowledge, such comparison of thermal expansion between different organic and inorganic PQDs, has never been reported so far. Here is our guess: as we know, the thermal expansion is strongly influenced by crystal phase, QD size, lattice strain, as well as chemical composition. To obtain a stable black-phase (photoactive) ABX_3 perovskite, the

Goldschmidt tolerance factor should not be lower than 0.8, nor exceed a value of 1 that indicates an ideal cubic phase. This tolerance value outside of this range usually results in non-perovskite structures (yellow phase). In the case of CsPbI₃, the tolerance factor is too small (0.83), whereas in the case of FAPbI₃ the tolerance factor is about 1. This results in the black-phase CsPbI₃ that cannot be easily stabilized at room temperature, not like FAPbI₃. However, surface ligands inducing strain stabilize the black phase of CsPbI₃ PQDs under ambient condition. In other words, the CsPbI₃ PQDs (or Cs-based PQDs) possess significant negative surface energy and lattice strain. Therefore, although it is typically believed that the thermal expansion coefficient of hybrid inorganic-organic FAPbI₃ are larger than all-inorganic CsPbI₃, the effect of surface and inter lattice strain in QD-form perovskite may overcome the influence of the chemical composition, resulting in that the CsPbI₃ PQDs have larger thermal expansion coefficient than FAPbI₃ PQDs. The descriptions have been added in the revised manuscript as “It can be seen that the linear thermal expansion coefficient increases with increasing the Cs content of PQDs. Although it is typically believed that the thermal expansion coefficient of hybrid inorganic-organic perovskites is larger than that of all-inorganic perovskites, this opposite trend for QD-form perovskite here is likely due to the significant negative surface energy and lattice strain in Cs-based PQDs.”

8. The author mentioned that thermal stabilities of all inorganic perovskite materials are higher than those of organic and organic-inorganic hybrid halide perovskites. However, in this research, it is revealed that FAPbI₃ PQDs showed the higher stabilities than those of CsFAPbI₃, Cs_xFA_{1-x}PbI₃ PQDs. Is this case only happening I₃ perovskite and/or FA perovskite QDs? Is this case also happening when Br or other A cation is used?

Response: Thanks for the comment. In this work, we demonstrate that the thermal stability of PQDs is not only dependent on chemical compositions, but also on their ligand binding, especially for Cs_xFA_{1-x}PbI₃ PQDs that we are focusing on here. When the ligand binding plays a key role on the stabilization of crystal phase such as in CsPbI₃, the influence of ligands would lead to this result that CsPbI₃ PQDs show poorer stability than FAPbI₃ PQDs. In case of Br- or another A cation-based PQDs, the thermal stability is associated with how the ligand binding functions. For example, CsPbBr₃ PQDs are intrinsically stable at room temperature (tolerance factor is 0.92) and the ligand binding is not essential for stabilizing their perovskite phase. Therefore, we expect that CsPbBr₃ PQDs may show higher stability than FAPbBr₃ PQDs. To further clarify the relationship between ligand binding energy, chemical position and phase stability on PQDs, we would like to see more research on these fundamental properties, and one or more related review papers could be published in the near future.

9. No Figure 3c in Text. And no explanation of some SI figures (For example S3-S6). Check it.

Response: Thanks, and sorry for the carelessness. The modifications are shown as “As illustrated in Fig. 4C, FA-rich PQDs exhibit more negative values of formation energy than Cs-rich PQDs, suggesting that the FA-rich PQDs having the perovskite phase with corner shared octahedra is more thermodynamically favorable at room temperature.”

The explanation of **Fig. S4-S8** was presented in the caption of the **Fig. 1**. We now move it in the main text as “The detailed analyses of *in situ* XRD profiles are demonstrated in Fig. S4-S8.” All figures have been checked in the revised manuscript.

10. Page 12, 298-300, “As illustrated in Fig. 3C, the calculation results of formation energies strictly for the perovskite phase with corner shared octahedra also explain why FA-rich PQDs have the higher thermal stability than Cs-rich PQDs.” The reviewer cannot understand scientific meaning of above sentence. Explain it in detail

Response: Thanks. The sentence is modified as “As illustrated in Fig. 4C, FA-rich PQDs exhibit more negative values of formation energy than Cs-rich PQDs, suggesting that the FA-rich PQDs having the perovskite phase with corner shared octahedra is more thermodynamically favorable at room temperature.”

11. Page 15,353, Not Figure 4C. Check it.

Response: Thanks. The Fig. 3C (there should not be Fig. 3C) is replaced by Fig. 4C in the revised manuscript, as “As illustrated in Fig. 4C, FA-rich PQDs exhibit more negative values of formation energy than Cs-rich PQDs, suggesting that the FA-rich PQDs having the perovskite phase with corner shared octahedra is more thermodynamically favorable at room temperature.”

Reviewer #4 (Remarks to the Author):

The manuscript examines the thermal degradation mechanism of the CsPbI₃, FAPbI₃ and mixed A-site composition of them. The results show that the thermal degradation processes are different for different compositions of PQDs. While the FA rich PQDs decomposes into PbI₂ at elevated temperatures, Cs rich PQDs has gone into a phase transition before decomposing into PbI₂. In the manuscript XRD measurements are studied to define the phase transitions, TGA measurements are investigated the degradation mechanisms in the presence of organic ligands and temperature dependent PL spectroscopy reveals the exciton-phonon interactions at elevated temperatures. Although the study is significant in understanding the fundamentals of thermal stress induced degradation processes in mixed A-site FA_xCs_{1-x}PbI₃ QDs, below points need to be addressed for a complete explanation of some results and discussions. Therefore, I recommend a major revision of this manuscript before publication.

Response: Thanks for your kind comment on this work. We have revised the manuscript with point-to-point response to your questions and suggestions.

1) Degradation mechanisms for the Cs- and FA-rich PQDs are studied in detail in comparison with their pure counterparts. However, I couldn't find enough discussion in the manuscript explaining the degradation mechanism for Cs_{0.5}FA_{0.5}PbI₃.

Response: Thank you for the pertinent suggestion. When Cs_xFA_{1-x}PbI₃ PQDs contain more Cs, the degradation process of PQDs is more dominated by phase transition from black perovskite to yellow non-perovskite phase. When the FA content is more than Cs content in PQDs, the degradation is mainly attributed to the decomposition into PbI₂. In the regard of Cs_{0.5}FA_{0.5}PbI₃ PQDs, the degradation process seems to be governed by the decomposition into PbI₂, while the phase transition to yellow non-perovskite phase have been also observed distinctly. We assume that these two trends are in competition to achieve this phenomenon (more like FA-rich PQDs). Following your suggestion, the discussion on Cs_{0.5}FA_{0.5}PbI₃ PQDs has been added in the revised manuscript as “In the case of Cs_{0.5}FA_{0.5}PbI₃ PQDs, both the δ-phase nanorods and hexagonal PbI₂ are clearly noticeable in SEM images. Combined with the XRD result presented in Fig. 1, it allows us to assume that the degradation process of Cs_{0.5}FA_{0.5}PbI₃ PQDs seems to be governed by the decomposition into PbI₂, while the phase transition to yellow non-perovskite phase also plays a big role. These two trends are in competition to degrade Cs_{0.5}FA_{0.5}PbI₃ PQDs.”

2) It is important to investigate the degradation mechanism of these mixed A-site PQDs. However the devices using these QDs will not going to be working at these elevated temperatures. For practical point of view, I find it useful to include the possible cooling down phase transformation of these PQDs after heated up to the phase transitions temperatures.

Response: Thanks for this comment. For FAPbI₃ PQDs, as increasing the heating temperature, some of PQDs are decomposed into PbI₂ and the other PQDs undergo

grain growth/merging. Thus, bulk FAPbI_3 with the degradation product PbI_2 can be found when cooling down. For CsPbI_3 PQDs, as we know, the non-perovskite δ -phase with wide band gap (E_g : ~ 2.82 eV, yellow) is thermodynamically more stable than perovskite black phase below ~ 320 °C in general. The reason that the black phase of CsPbI_3 PQD can be stabilized at room temperature is ligand binding to form a negative surface energy (*ACS Energy Lett.* **5**, 238–247 (2020)). Therefore, when the surface ligands are evaporated with increasing the temperature, the black phase of CsPbI_3 is transformed to yellow phase which is thermodynamically stable at room temperature. So, when the temperature is cooled down to room temperature, the CsPbI_3 still exhibit non-perovskite δ -phase, rather transformation to black phase. Following your suggestion, we carrier out the in situ XRD to examine the structural evolution of $\text{Cs}_x\text{FA}_{1-x}\text{PbI}_3$ PQDs when the temperature is cooled down from ~ 200 °C (phase transitions temperature or decomposition temperature) to room temperature.

Fig. S13. (A-E) In situ XRD experiments as the temperature is cooled down from phase transition or decomposition temperature to room temperature for CsPbI_3 (A), $\text{Cs}_{0.75}\text{FA}_{0.25}\text{PbI}_3$ (B), $\text{Cs}_{0.5}\text{FA}_{0.5}\text{PbI}_3$ (C), $\text{Cs}_{0.25}\text{FA}_{0.75}\text{PbI}_3$ (D), and FAPbI_3 (E) PQDs

It can be seen in **Fig. S13** that Cs-rich PQDs exhibit non-perovskite δ -phase and FA-rich (including $\text{Cs}_{0.5}\text{FA}_{0.5}\text{PbI}_3$) PQDs show stronger and narrower XRD peaks suggesting the bulk perovskites after the temperature is cooled down to room temperature. There is no further phase transformation and degradation during this cooling down process. The corresponding description is added in the revised manuscript as “Moreover, to broaden the understanding of the degradation mechanism of these PQDs, we further perform in situ XRD to show the structural evolution of $\text{Cs}_x\text{FA}_{1-x}\text{PbI}_3$ PQDs, when the temperature is cooled down from ~ 200 °C (phase transition or decomposition temperature) to room temperature, as demonstrated in **Fig. S13**. It can be seen that there is no further phase transformation and degradation during the cooling down process.”

3) The authors claim in the conclusion that FA-rich PQDs are favorable for PV applications based on the conclusion that photogenerated excitons are dissociating by LO photon scattering. However this claim is based on the fitting at elevated temperatures without considering ionized impurities. Also it is well known in the literature that ionized impurities has significant effect on the PV device performance. Considering the operating temperature range of PV devices, can the authors comment on the validity of this claim?

Response: Thanks for the comment and we are sorry for the confusion on the temperature unit. Since the temperature falls below 0 °C, we change the unit to **kelvin** in the description and figures of in situ PL spectra (**Fig.5**), as usually done in previous reports. Therefore, the parameters presented in Table S3 and S4 are obtained from the fitting in the temperature range of 80-400K that covers the operating temperature range of PV devices (~ 298 -353K or ~ 25 -85°C). In the regard of ionized impurities, this impurity term in the equation, $\Gamma(T) = \Gamma_0 + \Gamma_{\text{AC}} + \Gamma_{\text{LO}} + \Gamma_{\text{imp}}$, is the linewidth arising

due to scattering from fully ionized impurities which is governed by their average binding energy. The impurity contribution is almost negligible for impurity concentrations of below 10^{-15} cm^{-3} , or could be considered when the temperature is below 20K. In the case of iodide PQDs, LO phonon interaction typically dominates the PL broadening without the inclusion of the impurity term as the temperature is above 100 K (*Adv. Optical Mater.* **5**, 1700231 (2017); *Phys. Rev. Mater.***5**, 095402 (2021)). In another aspect, we believe that, without any constrain, less variables in the equation used for fitting such a simple curve in Fig. 5B, suggests more accuracy in the calculated results. As shown in the below **Figure R7** and **Table R1**, γ_{imp} is 1–2 orders of magnitude smaller than γ_{LO} , and we don't think that these values are reliable for further analysis. Thus, to avoid misleading conclusions when interpreting the impurity term, we would not suggest showing these fittings to the audience. To provide some information on device-related properties, we present PLQY and TRPL for different PQDs in **Fig. S1** (shown in Response to Review 3# Q1), indicating FA-rich PQDs having less non-radiative recombination than Cs-rich PQDs.

Figure R7. FWHM for FAPbI₃, Cs_{0.25}FA_{0.75}PbI₃, Cs_{0.5}FA_{0.5}PbI₃, Cs_{0.75}FA_{0.25}PbI₃, and CsPbI₃ PQDs. The black squares are the experimental data fitted using Segall’s expression (red solid line). The PL exciton linewidth broadening arises from inhomogeneous broadening term $\Gamma(T) = \Gamma_0$ (orange dash-dot line), acoustic phonon interaction $\Gamma(T) = \Gamma_0 + \Gamma_{AC}$ (blue dash-dot line), longitudinal optical phonon interaction $\Gamma(T) = \Gamma_0 + \Gamma_{LO}$ (green dash-dot line) and ionized impurity interaction $\Gamma(T) = \Gamma_0 + \Gamma_{imp}$ (shallow blue dash-dot line)

Table R1. Extracted PL linewidth parameters from fits of Eq.4 including Γ_0 , Γ_{AC} , Γ_{LO} , and Γ_{imp} terms for FAPbI₃, Cs_{0.25}FA_{0.75}PbI₃, Cs_{0.5}FA_{0.5}PbI₃, Cs_{0.75}FA_{0.25}PbI₃, and CsPbI₃ PQDs. The uncertainties noted are standard-errors from the fit procedure without including the systematic errors from the performed experiments.

PQDs	Γ_0 (meV)	γ_{AC} (meV)	E_{LO} (meV)	γ_{LO} (meV)	γ_{imp} (meV)
CsPbI ₃	60.0±0.31	0.019	71.4±2.35	342.0±27.61	12.04±2.3
Cs _{0.75} FA _{0.25} PbI ₃	62.4±0.22	0.055	106.1±2.63	778.5±65.7	8.00±0.5
Cs _{0.5} FA _{0.5} PbI ₃	66.3±0.28	0.048	114.7±6.10	984.5±200.2	9.48±1.5
Cs _{0.25} FA _{0.75} PbI ₃	69.3±0.20	0.043	115.2±3.02	1175.4±113.0	10.10±1.1
FAPbI ₃	74.4±0.18	0.015	125.8±6.17	1454.3±322.	14.37±3.4

4) In Fig 5a, temperature dependent PL spectra is given. Does the authors observe any significant change in the PL spectra at the phase transition temperatures? Also, it is shown in the figures that PL FWHM narrows at temperatures between 250°C and 300°C in all PQDs except FAPbI₃. Can the authors comment on this behaviour?

Response: Thank you for your valuable comment. Based on XRD results, the significant phase transformation from black to yellow phase for Cs-based PQDs occurs at above 400 K. In the temperature range from 80 K to 400 K, there is no discontinuity

in bandgap energy, but a S-shape behavior in **Fig. 5D**. As described in the paper, this S-shape behavior may be resulted from the significantly increased electron–LO phonon interaction in this temperature range, but also can be ascribed to the phase transition that is previously reported in perovskite bulks, demonstrating FA-based halide perovskite exhibit a continuous phase transition from the cubic α -phase to the tetragonal β -phase at ~ 285 K. It is difficult to accurately and precisely examine how each fraction affects PL emission in these $\text{Cs}_x\text{FA}_{1-x}\text{PbI}_3$ PQD systems based on the current experimental data. Many studies have also shown the high complexity in the temperature dependence of the band-gap energies for these PQDs is owing to the unique ligand binding on the soft perovskite lattice accompanied with quantum confinement, which is distinguished from their bulk counterparts.

In addition, we do observe an abrupt change in the PL intensity at ~ 250 K and ~ 330 K in **Fig. 5C**. The PL emission intensity from most of semiconductors is normally observed to decrease monotonically with increasing temperature, which is called PL thermal quenching that is basically due to the increase of the nonradiative recombination probability of electrons and holes. However, for these $\text{Cs}_x\text{FA}_{1-x}\text{PbI}_3$ PQD, the increase of PL intensity in the range of ~ 250 - 330 K is attributed to that the electrons captured by defects gain energy from collision or surrounding environment to escape from the defect level and participate in radiative recombination, which overcomes the influence of thermal quenching and gives rise to the PL intensity. It has been reported that the variation process is also dependent on the excitation power of PL measurements. The description is modified as “It can be seen in Fig. 5C that an abrupt change in the PL intensity occurs at ~ 250 K and ~ 330 K. The PL emission intensity for most of semiconductors is usually observed to decrease monotonically with increasing temperature (termed as thermal quenching), basically due to the increase of the nonradiative recombination probability of electrons and holes. However, for these $\text{Cs}_x\text{FA}_{1-x}\text{PbI}_3$ PQD, the increase of PL intensity in the range of ~ 250 - 330 K is attributed to that the electrons captured by defects gain energy from collisions or surroundings to escape from the defect cavity and participate in radiative recombination, overcoming the influence of the thermal quenching and giving rise to the PL intensity. It has been reported that this variation process in PL intensity is also dependent on the excitation power of PL measurements.”

As for PL FWHM narrow, we were in same image that there was an apparent narrowing in FWHM in the range of ~ 250 - 330 K, when we just plotted the raw data of in situ PL spectra in **Fig. 5A**. However, as we analyse the FWHM shown in **Fig. 5B**, it can be found in very detail that only a little change in FWHM for all PQDs occurred in the range of ~ 250 - 330 K, which may be influenced by the variation of PL intensity as described above. Therefore, the PL FWHM narrowing seems to be a visual illusion, because the red colour is obviously narrowed (the PL intensity is decreased) while the whole width of PL peak with green and/or shallow green is actually not changed. This is also possibly resulted from the normalization of PL intensity when producing the pseudo colour image. We apologize for this visual illusion and plotted the raw data only using two colours. As can be seen in the following **Figure R8**, the visual illusion is mostly eliminated but many details (such as the change of intensity) are lost. So, we

add an explanation in the revised manuscript as “In addition, the width narrowing in red colour during the temperature range of ~250-330K (Fig. 5A) is a visual illusion of the decrease in PL FWHM which is actually caused by the decrease of PL intensity.”

Figure R8. In situ PL spectra for FAPbI₃, Cs_{0.25}FA_{0.75}PbI₃, Cs_{0.5}FA_{0.5}PbI₃, Cs_{0.75}FA_{0.25}PbI₃, and CsPbI₃ PQDs.

REVIEWER COMMENTS

Reviewer #1 (Remarks to the Author):

I have read through all the comments and the point-by-point responses. The authors' replies have well addressed the comments. I only have simple further suggestion to authors:

In Page 8, for the opposite trends in PQDs and bulk counterparts about the thermal stability. Author claims that “These unique and opposite trends in PQDs must be attributed to a synergistic effect of chemical composition and surface energy induced by ligand binding, which is the key to stabilize black perovskite phase at room temperature especially for Cs-alloyed perovskite nanoparticles.” I agree this explanation but I also think it is very important and can even be one of the core points of the article. Author have only proposed and simply stated the “synergistic effect” , but I think the theoretical depth is still shallow. For example, what is the “synergistic effect”? and how do chemical composition and surface energy separately and jointly influence phase stability? What’s the mechanism or And I suggest author should provide some experimental data or theoretical calculations for explaining this “synergistic effect”. This comment is not to criticize but to enhance the manuscript.

Reviewer #2 (Remarks to the Author):

After carefully reviewing the revised manuscript and the rebuttal letter, the authors have addressed all questions raised by the reviewer. The reviewer thinks this manuscript is ready to be published.

Reviewer #4 (Remarks to the Author):

I would like to thank the authors for their detailed responses to the questions. I think the degradation mechanisms are clear now and the manuscript is strong enough to be publish at this time.

Response to Reviewer Comments

Reviewer #1 (Remarks to the Author):

I have read through all the comments and the point-by-point responses. The authors' replies have well addressed the comments. I only have simple further suggestion to authors:

In Page 8, for the opposite trends in PQDs and bulk counterparts about the thermal stability. Author claims that “These unique and opposite trends in PQDs must be attributed to a synergistic effect of chemical composition and surface energy induced by ligand binding, which is the key to stabilize black perovskite phase at room temperature especially for Cs-alloyed perovskite nanoparticles.” I agree this explanation but I also think it is very important and can even be one of the core points of the article. Author have only proposed and simply stated the “synergistic effect”, but I think the theoretical depth is still shallow. For example, what is the “synergistic effect”? and how do chemical composition and surface energy separately and jointly influence phase stability? What's the mechanism or And I suggest author should provide some experimental data or theoretical calculations for explaining this “synergistic effect”. This comment is not to criticize but to enhance the manuscript.

Response: Thank you for your kind comment and valuable suggestion. Generally, Goldschmidt's tolerance factor was used in researches to rationalize the compositionally dependent formability of perovskites semi-quantitatively. The tolerance factor (τ) can be calculated by the following formula: $\tau = (r_A+r_X)/(\sqrt{2}(r_B+r_X))$; In addition, the formability (μ) of the octahedra is determined by: $\mu = r_B/r_X$, where r represents the effective radii of A, B, and X ions in perovskite formula (ABX_3), respectively. The black 3D perovskite phase is stable when $0.813 \leq \tau \leq 1.107$ and $0.442 \leq \mu \leq 0.895$. That's how the chemical composition influences phase stability directly. For perovskite quantum dots, especially for Cs-rich or pure-Cs PQDs, the ligand binding that induces negative surface energy plays a key role to stabilize the black phase particularly when this black phase is thermodynamically unstable at room temperature. In this work, we demonstrate that the ligand binding actually have strong correlations not only with the types of surface ligand, but also the chemical composition in perovskite crystals. These two factors concurrently have significant effect on the phase stability, and also impact each other. Therefore, we stated that “These unique and opposite trends in PQDs must be attributed to a synergistic effect of chemical composition and surface energy induced by ligand binding, which is the key to stabilize black perovskite phase at room temperature especially for Cs-alloyed perovskite nanoparticles.” Following your suggestions, we prepare $Cs_{0.75}FA_{0.25}PbI_3$ PQDs with trioctylphosphine oxide (TOPO) as surface binding ligands, and perform XRD experiments as well as DFT theoretical calculations on them. The $Cs_{0.75}FA_{0.25}PbI_3$ PQDs with TOPO ligands exhibit better thermal stability than the ones with OA and OAm ligands. The binding energy of TOPO ligands on $Cs_{0.75}FA_{0.25}PbI_3$ PQDs (-1.15 eV) is higher than that of OA (-0.55 eV) and OAm (-0.54 eV) ligands. These suggest that the different ligands on the same kind of

PQDs provide different thermal stabilities, through changing the binding between the ligands and the PQD surface. Combined with the results from the above XRD and DFT analyses on PQDs with same surface ligands but different chemical compositions, it implies that the phase stability is synergistically influenced by the chemical composition and surface ligands. The manuscript has been revised as “**To further clarify the relation between the chemical composition and surface ligand binding, and how it impacts the thermal stability, we prepare Cs_{0.75}FA_{0.25}PbI₃ PQDs with trioctylphosphine oxide (TOPO) as surface binding ligands, and perform XRD experiments as well as DFT theoretical calculations on them. In Figure S16-S18, the Cs_{0.75}FA_{0.25}PbI₃ PQDs with TOPO ligands exhibit better thermal stability than the ones with OA and OAm ligands. The binding energy of TOPO ligands on Cs_{0.75}FA_{0.25}PbI₃ PQDs (-1.15 eV) is higher than that of OA (-0.55 eV) and OAm (-0.54 eV) ligands. These suggest that the different ligands on the same kind of PQDs provide different thermal stabilities, through changing the binding between the ligands and the PQD surface. Combined with the results from the above XRD and DFT analyses on PQDs with same surface ligands but different chemical compositions, it implies that the phase stability is synergistically influenced by the chemical composition and surface ligands.**”

Figure S16. (A) *In situ* XRD patterns collected from 30 °C to 500 °C under argon flowing for Cs_{0.75}FA_{0.25}PbI₃PQDs with (right panel) and without (left panel) TOPO ligands. (B) Scheme for the dominant reflections in the thermal degradation of Cs_{0.75}FA_{0.25}PbI₃ PQDs with (below) and without (upper) TOPO ligands that is drawn from the vertical line cuts of the 2-dimensional *in situ* XRD patterns taken at the 2θ

angles shown in Table S1 orthorhombic non-perovskite (non-PVK yellow phase) and PbI_2 .

Figure S17. (A-B) Ex situ XRD patterns of $\text{Cs}_{0.75}\text{FA}_{0.25}\text{PbI}_3$ PQDs without (A) and with (B) TOPO ligands. (C) Photographs of $\text{Cs}_{0.75}\text{FA}_{0.25}\text{PbI}_3$ PQD films with and without TOPO ligands at different temperatures.

Figure S18. (A-C) Charge density difference plot for one OA (A), one OAm (B) or one TOPO (C) ligand adsorbed on the surface of $\text{Cs}_x\text{FA}_{1-x}\text{PbI}_3$ PQDs. Red and green denotes areas of electron accumulation and depletion in electron localization function analysis, respectively.

Reviewer #2 (Remarks to the Author):

After carefully reviewing the revised manuscript and the rebuttal letter, the authors have addressed all questions raised by the reviewer. The reviewer thinks this manuscript is ready to be published.

Response: Thanks for the comment.

Reviewer #4 (Remarks to the Author):

I would like to thank the authors for their detailed responses to the questions. I think the degradation mechanisms are clear now and the manuscript is strong enough to be publish at this time.

Response: Thanks for the comment.